# PI3P-dependent regulation of cell size and autophagy by phosphatidylinositol 5-phosphate 4-kinase

Avishek Ghosh, Aishwarya Venugopal, Dhananjay Shinde, Sanjeev Sharma, Meera Krishnan, Swarna Mathre, Harini Krishnan, Sankhanil Saha, Padinjat Raghu

Phosphatidylinositol 3-phosphate (PI3P) and phosphatidylinositol 5-phosphate (PI5P) are low-abundance phosphoinositides crucial for key cellular events such as endosomal trafficking and autophagy. Phosphatidylinositol 5-phosphate 4-kinase (PIP4K) is an enzyme that regulates PI5P in vivo but can act on both PI5P and PI3P in vitro. In this study, we report a role for PIP4K in regulating PI3P levels in *Drosophila*. Loss-of-function mutants of the only *Drosophila* PIP4K gene show reduced cell size in salivary glands. PI3P levels are elevated in *dPIP4K²⁹* and reverting PI3P levels back towards WT, without changes in PI5P levels, can rescue the reduced cell size. *dPIP4K²⁹* mutants also show up-regulation in autophagy and the reduced cell size can be reverted by depleting Atg8a that is required for autophagy. Lastly, increasing PI3P levels in WT can phenocopy the reduction in cell size and associated autophagy up-regulation seen in *dPIP4K²⁹*. Thus, our study reports a role for a PIP4K-regulated PI3P pool in the control of autophagy and cell size.

## Introduction

The organization of membranes in eukaryotic cells is regulated by signalling mechanisms that couple ongoing stimuli to subcellular transport mechanisms. Several signalling molecules contribute to this process including proteins such as SNAREs and Rabs along with lipids. Phosphoinositides are a class of signalling lipids found in all eukaryotes; they are glycerophospholipids whose inositol head-group can be phosphorylated on the third, fourth or fifth positions to generate molecules with signalling functions (Balla, 2013). In cells, phosphoinositides are generated by the action of lipid kinases that are able to add phosphate groups to specific positions on the inositol head group of specific substrates (Sasaki et al, 2009); thus, the activity of these lipid kinases and phosphatases is important to generate lipid signals on organelle membranes. Phosphatidylinositol 5 phosphate 4-kinase (PIP4K) are one such class of lipid kinases that convert phosphatidylinositol 5 phosphate (PI5P)

into phosphatidylinositol 4,5 bisphosphate [PI(4,5)P$_2$] (Rameh et al, 1997; Clarke & Irvine, 2013). Genetic analysis of *PIP4K* in various organisms have demonstrated their importance in development and growth control (Gupta et al, 2013), cell division (Emerling et al, 2013), immune cell function (Shim et al, 2016), metabolism (Lamia et al, 2004), and neurological disorders (Al-Ramahi et al, 2017). At a cellular level, PIP4K have been implicated in the control of plasma membrane receptor signalling (Sharma et al, 2019), vesicular transport (Kamalesh et al, 2017), autophagy (Vicinanza et al, 2015; Lundquist et al, 2018), and nuclear functions such as transcriptional control (Fiume et al, 2019). PI(4,5)P$_2$, the product of PIP4K activity has many important functions in regulating cell physiology and signalling (Kolay et al, 2016) and PI5P, the well-defined substrate of PIP4K has also been implicated in regulating some subcellular processes (Hasegawa et al, 2017). However, despite their importance in regulating several cellular processes and physiology, the biochemical reason for the requirement of PIP4K in regulating these processes remain unclear.

When studied using biochemical activity assays in vitro, PIP4K shows very high activity on PI5P to generate PI(4,5)P$_2$ (Rameh et al, 1997; Zhang et al, 1997; Ghosh et al, 2019). Coupled with this, analysis of lipid levels following genetic depletion of PIP4K in various models have failed to note appreciable reductions in PI(4,5)P$_2$ levels (reviewed in Kolay et al [2016]). Rather, such studies have reported an increase in the levels of the substrate PI5P (Jones et al, 2006; Gupta et al, 2013; Stijf-Bultsma et al, 2015) suggesting that the relevant biochemical function of the enzyme is to regulate PI5P levels. Previous studies have noted that PIP4K depletion in *Drosophila* photoreceptors (Kamalesh et al, 2017) leads to altered endocytic function and a role for PI5P in regulating endocytosis has been proposed (Ramel et al, 2011; Boal et al, 2015). In mammalian cells, PI5P has been proposed as a mediator of autophagy regulation by PIP4K (Vicinanza et al, 2015; Al-Ramahi et al, 2017; Lundquist et al, 2018). PIP4K can also use PI3P as a substrate in vitro (Zhang et al, 1997; Gupta et al, 2013; Ghosh et al, 2019), albeit with low efficiency; however, the significance of this activity in vivo and the role of PIP4K, if any, in regulating PI3P levels in vivo is not known. PI3P is well known as a regulator of autophagy (Schink et al, 2016; Wallroth & Haucke, 2018), a process that is reported to be altered on

National Centre for Biological Sciences, TIFR-GKVK Campus, Bangalore, India

Correspondence: praghu@ncbs.res.in

modulating PIP4K function but the significance, if any, of PIP4K-regulated pools of PI3P in these processes remains unknown. PI3P formed at the phagophore membrane by Vps34, a class III PI3-kinase is important for autophagy initiation by recruiting proteins like DFCP1, WIPI (Axe et al, 2008; Polson et al, 2010). In the next step, the ATG16L1 complex, which includes the proteins ATG16L1, ATG5, and ATG12, is recruited to the pre-autophagosomal membranes (Dudley et al, 2019) and Myotubularins, 3-phosphatase enzymes that have been reported to regulate autophagy by regulating PI3P levels at autophagy initiation membranes (Vergne et al, 2009; Taguchi-Atarashi et al, 2010; Zou et al, 2012). This raises the possibility that the reported regulation of autophagy by PIP4K may arise from its ability to regulate PI3P levels at the autophagic membrane.

The *Drosophila* genome contains a single gene encoding PIP4K (*dPIP4K*). A loss-of-function allele of *dPIP4K* (*dPIP4K^29*) results in altered growth and development, accumulation of the known substrate PI5P, and no reduction in PI(4,5)P$_2$ levels (Gupta et al, 2013). In *dPIP4K^29*, the size of larval salivary gland cells is reduced and genetic reconstitution studies have demonstrated that the kinase activity of dPIP4K is required to support normal cell size (Mathre et al, 2019). Previous work has shown that TORC1 signalling, a known regulator of cell size (Lloyd, 2013) and autophagy (Nascimbeni et al, 2017), is reduced in *dPIP4K^29* (Gupta et al, 2013). Thus, although it is clear that the kinase activity of dPIP4K is required for normal salivary gland cell size, the biochemical basis for this requirement for enzyme activity is not known.

In this study, we show that in addition to the previously reported elevation of PI5P levels, PI3P levels are also elevated in *dPIP4K^29* and this elevation in PI3P is dependent on the kinase activity of dPIP4K. The reduced salivary gland cell size in *dPIP4K^29* could be rescued by the expression of a PI3P-specific 3-phosphatase, *Mtm*, and this rescue was associated with a reversal of the elevated PI3P but not PI5P levels. Interestingly, we observed that in larval salivary glands of *dPIP4K^29*, the elevation in PI3P levels was associated with an up-regulation in autophagy and the phenotype of reduced cell size in *dPIP4K^29* could be reversed by down-regulating Atg8a, which functions downstream to the formation of PI3P in the autophagy pathway. Elevation of PI3P levels in WT salivary glands by depletion of *Mtm* resulted in both a reduction in cell size and the enhanced autophagy in salivary glands. Therefore, this study underscores a novel in vivo regulation of PI3P levels by PIP4K in a multicellular organism leading to the control of cell size.

# Results

### *dPIP4K* does not regulate cell size through levels of its product PI(4,5)P$_2$

The kinase activity of dPIP4K is required for its ability to support salivary gland cell size (Fig 1A depicts the conversion of PI5P to PI(4,5)P$_2$ by dPIP4K) (Mathre et al, 2019). Thus, its ability to regulate cell size may depend either on the elevated levels of its preferred substrate PI5P or a shortfall in the pool of the product PI(4,5)P$_2$ generated. Previous studies have identified a point mutation (A381E) in PIP4Kβ that can switch its substrate specificity from PI5P to PI4P

while still generating the same product PI(4,5)P$_2$ (Kunz et al, 2002). The corresponding point mutant version of hPIP4Kα has been used to distinguish between phenotypes dependent on the PI(4,5)P$_2$ generated by PIP4K as opposed to PI5P metabolised by it (Bulley et al, 2016). We generated a switch mutant version of human PIP4Kβ, hPIP4Kβ^[A381E] that cannot utilise PI5P as a substrate but can produce PI(4,5)P$_2$ using PI4P as a substrate (Kunz et al, 2002). Expression of hPIP4Kβ^[A381E] in the salivary glands of *dPIP4K^29* (*AB1> hPIP4Kβ^[A381E]; dPIP4K^29*) (Fig S1A) did not rescue the reduced cell size, whereas reconstitution with the WT enzyme was able to do so as previously reported (Fig S1B(i), quantified in Fig S1B(ii), Western blot in Fig S1A) (Mathre et al, 2019). This observation suggests that the ability of dPIP4K to regulate cell size does not depend on the pool of PI(4,5)P$_2$ that it generates and also that regulation of the levels of its substrate is likely to be the relevant biochemical basis through which the enzyme supports cell size in salivary glands.

### *Mtm* could be a candidate gene to modulate PI5P levels in *Drosophila*

Because PI5P is the preferred substrate of dPIP4K (Gupta et al, 2013), we sought to modulate PI5P levels to assess the impact on cell size regulation. However, other biochemical players involved in PI5P regulation in *Drosophila* are unknown so far. In mammals, PI5P levels are regulated by PIKFYVE, the type III PIP 5-kinase that converts PI3P to PI(3,5)P$_2$ and PI to PI5P (Shisheva, 2013; Hasegawa et al, 2017). *Drosophila* has a single PIKFYVE homologue (*CG6355*, here named *dFab1*) (Rusten et al, 2006); however, its biochemical activity has not been tested (Fig 1A). We expressed mCherry-tagged dFab1 in S2R+ cells, immunoprecipitated it (Fig 1C(i)) and analysed its ability to phosphorylate PI3P and PI, using an LC-MS/MS-based in vitro kinase activity assay for dFab1 (Fig 1B shows a schematic for the assay). We found that the relative activity of dFab1 on synthetic PI3P was ~4 times greater than the activity on synthetic PI (Fig 1C(ii)). Because dFab1 preferentially synthesizes PI(3,5)P$_2$ from PI3P, subsequent PI5P generation would require the activity of a 3-phosphatase that can dephosphorylate PI(3,5)P$_2$. In mammals, in vitro studies have revealed that lipid phosphatases of the myotubularin family have specific activity toward PI3P and PI(3,5)P$_2$ (Fig 1A) (Laporte et al, 1996; Walker et al, 2001; Schaletzky et al, 2003). In most higher order organisms, there are multiple myotubularin isoforms (Robinson & Dixon, 2006). It has been suggested that *Drosophila* has six isoforms (Oppelt et al, 2013), but bioinformatics analysis using multiple sequence alignment revealed that the conserved CX$_5$R catalytic motif is present in only three genes – *Mtm*, *CG3632*, and *CG3530* (Fig S1C). To identify the myotubularin that might generate PI5P from PI(3,5)P$_2$, we designed a two-step in vitro LC-MS/MS-based PI(3,5)P$_2$ 3-phosphatase assay using *Drosophila* S2R+ cell lysates as a source of enzyme (Fig 1D, details of the assay is mentioned in methods). Briefly, deuterium-labelled PI(3,5)P$_2$ [d5-PI(3,5)P$_2$] is incubated with cell lysate and the PI5P formed through the action of a 3-phosphatase is converted, using $^{18}$O-ATP to PI(4,5)P$_2$ of an unique mass owing to the incorporated $^{18}$O, and subsequently detected on a mass spectrometer (Fig 1D). We used a linked PI5P-4-kinase assay to distinguish a 3-phosphatase activity generating PI5P from a 5-phosphatase activity generating PI3P, from the cell lysates in the first step of the assay. Each of the 3-phosphatases (*Mtm, CG3632,*

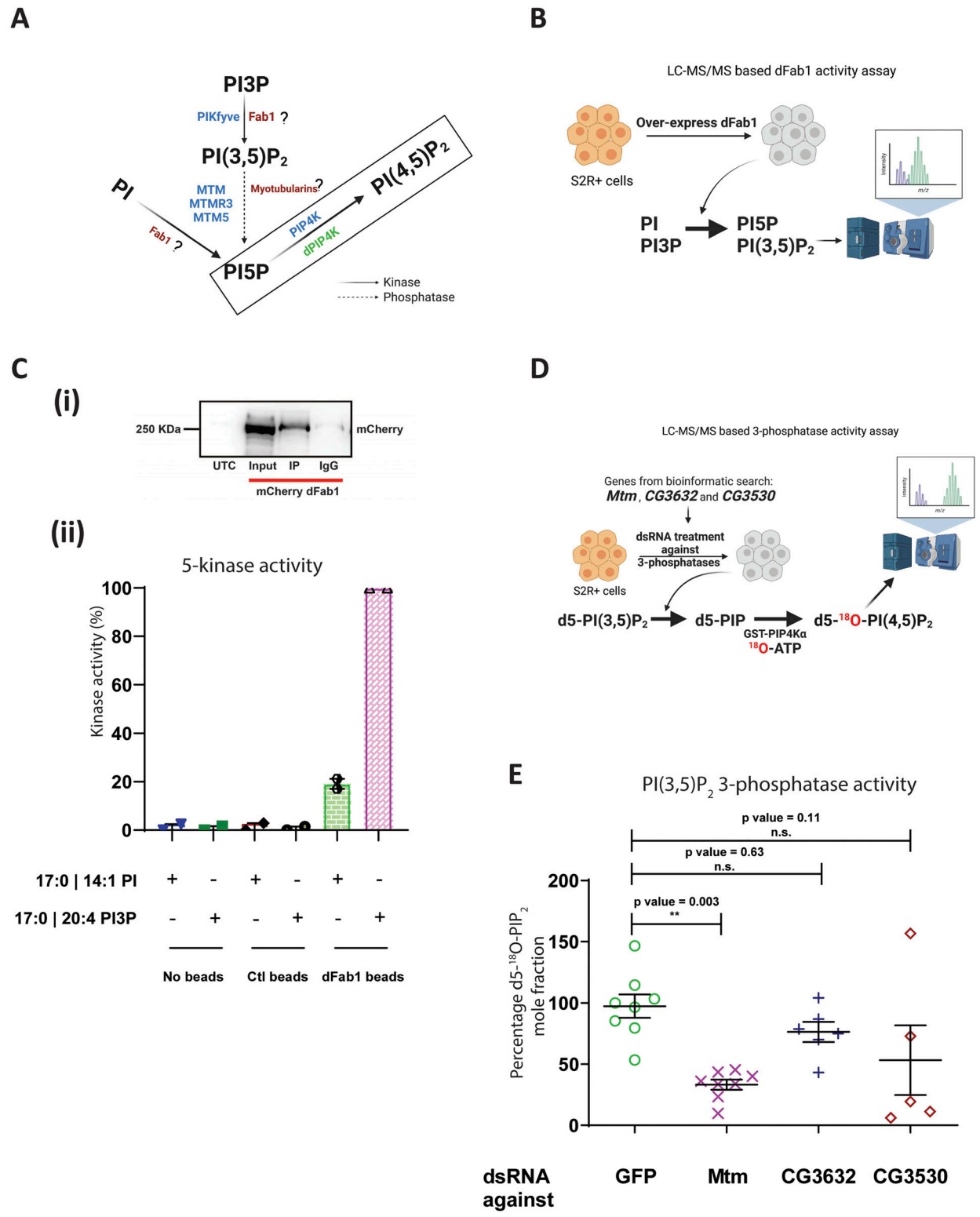

**Figure 1. Screening for a biochemical route to modulate PI5P levels in *Drosophila* as altering local PI(4,5)P$_2$ could not change the cell size of *dPIP4K$^{29}$*.**
**(A)** Schematic illustrating the putative enzymatic routes by which PI5P can be synthesised in *Drosophila*. The activities of enzymes labelled in blue are known in mammalian cells, the activity of enzymes labelled in red followed by "?" are still unknown in *Drosophila*, the activity of enzymes labelled in green are known in *Drosophila*. The activity of PI5P to PI(4,5)P$_2$ is boxed and is linked to cell size regulation in *Drosophila* (Mathre et al, 2019). PI, phosphatidylinositol; PI3P, phosphatidylinositol 3-phosphate; PI(3,5)P$_2$, phosphatidylinositol 3,5 bisphosphate; PI5P, phosphatidylinositol 5-phosphate; PI(4,5)P$_2$, phosphatidylinositol 4,5 bisphosphate. Kinase and phosphatase activities are denoted by black solid arrows and black dashed arrows, respectively. **(B)** Schematic illustrating the LC-MS/MS-based in vitro 5-kinase activity

and *CG3530*) were depleted using dsRNA treatment (Worby et al, 2001) and the 3′ phosphatase activity of the lysates was measured. We noted more than 50% knockdown using dsRNA against *Mtm*, *CG3632,* and *CG3530* in S2R+ cells (Fig S1D(i–iii)). We observed that the d5-$^{18}$O-PIP$_2$ mole fraction (the measure of PI(3,5)P$_2$ 3-phosphatase activity) for *Mtm* down-regulated lysates was significantly lower as compared with control GFP dsRNA treated lysates (Fig 1E). However, we did not observe a significant difference in the activity of lysates down-regulated for *CG3632* or *CG3530*. Therefore, Mtm is a 3-phosphatase that could regulate PI5P synthesis in *Drosophila*.

### *Drosophila* Mtm reverses the cell size defect of *dPIP4K*$^{29}$ independent of PI5P levels

Based on the results of our in vitro experiments, we overexpressed *Mtm* in *dPIP4K*$^{29}$ salivary glands to elevate PI5P levels. If the reduced cell size in *dPIP4K*$^{29}$ was linked to elevated PI5P levels, *Mtm* overexpression in *dPIP4K*$^{29}$ is expected to lead to a further reduction in cell size. Surprisingly, we observed that overexpression of *Mtm* in the salivary glands of *dPIP4K*$^{29}$ (*AB1> Mtm*$^{WT}$::*GFP*; *dPIP4K*$^{29}$) resulted in a reversal of cell size as compared with *dPIP4K*$^{29}$ glands (Fig 2A(i), quantified in Fig 2A(ii)); overexpression of the enzyme in WT salivary glands did not affect cell size (Fig S2A).

Mtm is a 3-phosphatase that can act on PI3P to produce PI and PI(3,5)P$_2$ to produce PI5P. Previously, its activity on PI(3,5)P$_2$ has been demonstrated using purified protein in an in vitro phosphate-release assay (Velichkova et al, 2010). To understand the biochemical basis of the ability of overexpressed Mtm to reverse cell size in *dPIP4K*$^{29}$, we tested the biochemical activity of Mtm from *Drosophila* larval extracts using our two-step 3-phosphatase activity assay. Fig 2B shows the expression of C-terminus GFP-tagged Mtm from larval lysates at molecular weights as predicted in silico. We found that overexpression of Mtm did not result in a statistically significant increase in 3-phosphatase activity compared with controls (Fig 2C). To confirm this result was not an outcome of C-terminal tagging leading to Mtm inactivation, we cloned an N-terminus mCherry-tagged Mtm and performed the 3-phosphatase assay using S2R+ cell lysates expressing mCherry::Mtm (Fig S2B(i)). It was observed that an N-terminally tagged Mtm was also not active on PI(3,5)P$_2$ as compared with controls, much like its C-terminal GFP-tagged counterpart (Fig S2B(ii)). These findings suggest that the generation of PI5P from PI(3,5)P$_2$ by Mtm in *Drosophila* larvae is likely to be minimal. We also measured the levels of PI5P from larval lipid extracts using a recently standardised LC-MS/MS-based PI5P mass assay (Ghosh et al, 2019), comparing larvae expressing Mtm in

*dPIP4K*$^{29}$ mutant background with *dPIP4K*$^{29}$. We observed that the overexpression of Mtm did not alter the levels of PI5P in *dPIP4K*$^{29}$ (Fig 2D). Therefore, together, we conclude that (a) Mtm cannot synthesise PI5P from PI(3,5)P$_2$ in vivo in *Drosophila* and (b) Mtm expression rescued the cell size of *dPIP4K*$^{29}$ without changing the elevated PI5P levels. Therefore, we investigated PI5P-independent mechanism that could control cell size.

### Mtm reduces PI3P levels when overexpressed in *dPIP4K*$^{29}$

Mtm has also been shown to dephosphorylate PI3P to generate PI in vitro (Velichkova et al, 2010). We tested the activity of lysates expressing Mtm on synthetic PI3P using an LC-MS/MS-based assay and found that lysates with Mtm overexpression showed significantly higher PI3P 3-phosphatase activity compared with control lysates (Fig 3A), raising the possibility that Mtm expression might be rescuing cell size in *dPIP4K*$^{29}$ through its PI3P 3-phosphatase activity.

Mtm activity can in principle change the levels of PI and PI3P; however, because PI3P levels in cells are substantially lower (<10%) of PI (Stephens et al, 1993), we analysed PI3P levels in relation to the ability of Mtm overexpression to rescue the reduced cell size in *dPIP4K*$^{29}$ salivary glands. Currently used methods to quantify PI3P levels rely on radionuclide labelling techniques (Chicanne et al, 2012). We optimised a previously used label-free LC-MS/MS-based method to quantify PI3P levels from *Drosophila* larval lysates (Fig 3B depicts a chromatogram derived from injecting WT deacylated lipid samples) that allows the chromatographic separation and quantification of PI3P levels (Kiefer et al, 2010). To test if the ability of Mtm to dephosphorylate PI3P might be linked to its ability to reverse cell size in *dPIP4K*$^{29}$, we measured PI3P levels in these genotypes. We observed that PI3P was significantly reduced in *dPIP4K*$^{29}$ larvae expressing Mtm compared with *dPIP4K*$^{29}$ (Fig 3C). We also measured PI3P levels from larvae expressing Mtm in an otherwise WT background (Fig 3D) and found a modest reduction in the levels of PI3P. These results highlight the potential for PI3P levels to be correlated to the phenotype of cell size regulation in *Drosophila* salivary glands.

### dPIP4K regulates PI3P levels in vivo

Because reducing PI3P levels was correlated with cell size reversal in *dPIP4K*$^{29}$ (Figs 2A and 3C), we measured PI3P levels in *dPIP4K*$^{29}$. Interestingly, we observed that PI3P was elevated in *dPIP4K*$^{29}$ larvae as compared with controls (Fig 4A). To confirm this observation of elevated PI3P levels in *dPIP4K*$^{29}$ by an independent method, we

assay using S2R+ cells overexpressing dFab1 enzyme to convert synthetic PI or PI3P to PI5P and PI(3,5)P$_2$, respectively. **(C)** (i) Immunoprecipitated protein levels were analysed by Western blotting with an anti-mCherry antibody. Control (IgG) was prepared without anti-mCherry. Input lane shows the correct size of dFab1 protein ~230 kD. UTC, Untransfected control. (ii) In vitro kinase assay on synthetic PI and PI3P. Graph representing the kinase activity (%) as the normalised response ratio of "PI 5-kinase activity on PI" to "PI3P 5-kinase activity on PI3P" upon enzymatic activity of immunoprecipitated mCherry::dFab1 on the respective substrates. Response ratio of PI 5-kinase activity on PI is obtained from area under the curve of 17:0 14:1 PI5P (product)/17:0 14:1 PI (substrate), Response ratio of PI3P 5-kinase activity on PI3P is obtained from area under the curve of 17:0 20:4 PI(3,5)P$_2$ (product)/17:0 20:4 PI3P (substrate) and is represented as mean ± S.E.M. on addition of either negative control (no beads), control (mCherry beads) or dFab1 (mCherry::dFab1 beads). Number of immunoprecipitated samples = 2. **(D)** Schematic illustrating the LC-MS/MS-based in vitro PI(3,5)P$_2$ 3-phosphatase activity assay using dsRNA-treated S2R+ cells as the enzyme source to convert synthetic PI(3,5)P$_2$ [d5-PI(3,5)P$_2$ to d5-$^{18}$O- PIP$_2$] using a two-step reaction scheme. **(E)** In vitro phosphatase assay on synthetic PI(3,5)P$_2$. Graph representing the 3-phosphatase activity (%) as the percent formation of d5-$^{18}$O-PIP$_2$ formed from starting d5-PI(3,5)P$_2$ as mean ± S.E.M. on addition of either control (GFP ds RNA) or *Mtm, CG3632, CG3530* ds RNA-treated S2R+ cell lysates. One-way ANOVA with a post hoc Tukey's test shows *P*-value = 0.003 between GFP and *Mtm* ds RNA-treated lysates, shows *P*-value = 0.63 between *GFP* and *CG3632* ds RNA-treated lysate and shows *P*-value = 0.11 between *GFP* and *CG3530* ds RNA-treated lysates.

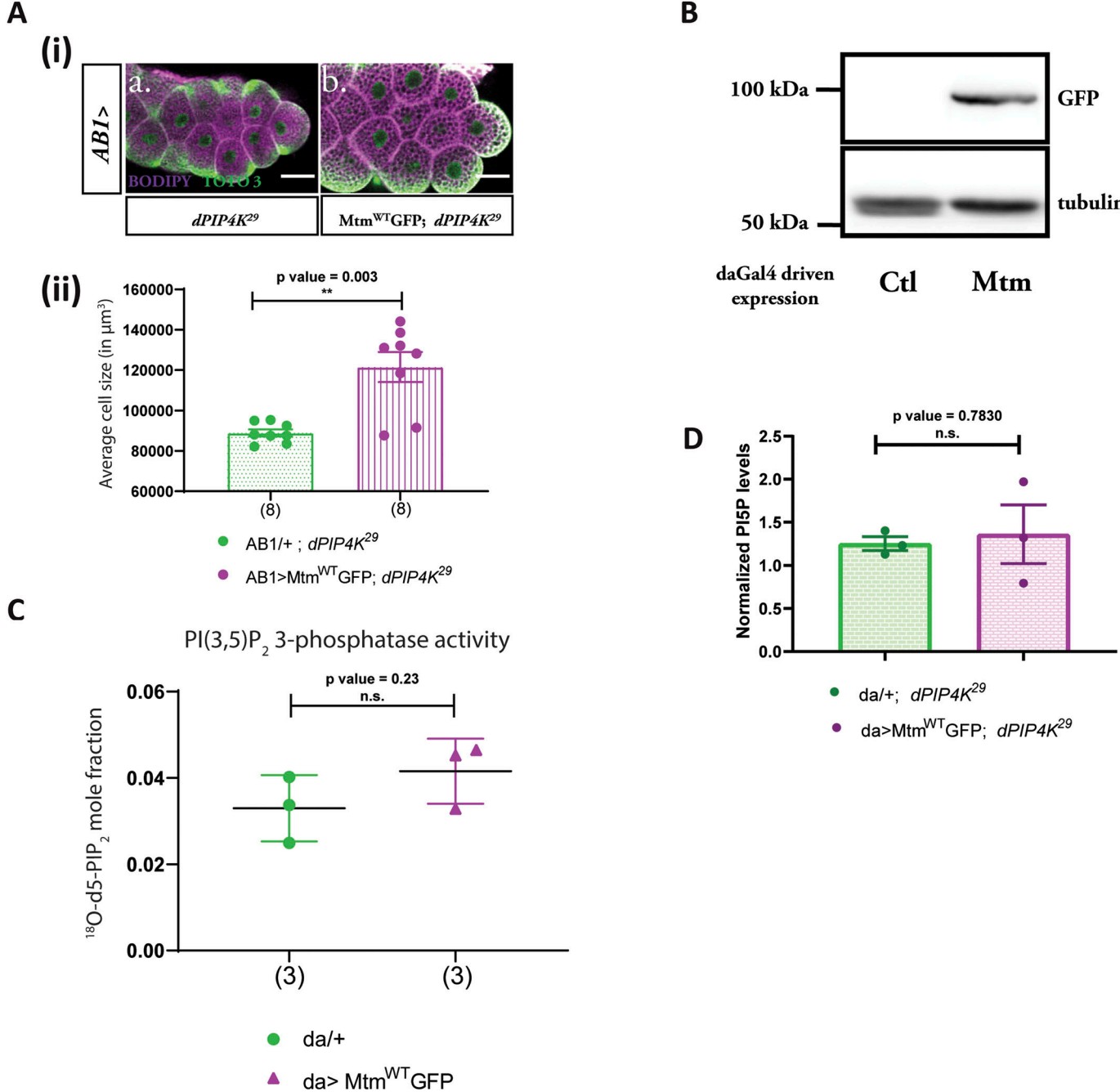

**Figure 2. *Drosophila* Mtm rescues the cell size defect of *dPIP4K²⁹* independent of PI5P levels.**
**(A)** (i) Representative confocal images of salivary glands from the genotypes a. *AB1/+; dPIP4K²⁹*, b. *AB1>Mtm^WT::GFP; dPIP4K²⁹*. Cell body is marked magenta by BODIPY conjugated lipid dye, nucleus is marked by TOTO-3 shown in green. Scale bar indicated at 50 $\mu$m. (ii) Graph representing average cell size measurement as mean ± S.E.M. of salivary glands from wandering third instar larvae of *AB1/+; dPIP4K²⁹* (n = 8), *AB1>Mtm^WT::GFP; dPIP4K²⁹* (n = 8). Sample size is represented on individual bars. Unpaired *t* test with Welch correction showed *P*-value = 0.003 between *AB1/+; dPIP4K²⁹* and *AB1>Mtm^WT::GFP; dPIP4K²⁹*. **(B)** Protein levels between *da/+* (Ctl) and *da> Mtm^WT::GFP* from third instar wandering larvae seen on a Western blot probed by GFP antibody. Mtm^WT GFP migrates ~100 kD. Tubulin was used as the loading control. **(C)** In vitro phosphatase assay on synthetic PI(3,5)P$_2$. Graph representing the formation of ¹⁸O-PIP$_2$ formed from starting PI(3,5)P$_2$ as substrate represented as mean ± S.E.M. on addition of either control (*da/+*) or *da>Mtm^WT::GFP* lysates. Lysate samples n = 3, where each sample was made from five third instar wandering larvae. Unpaired *t* test with Welch correction showed *P*-value = 0.23. **(D)** Graph representing normalised PI5P levels which is total ¹⁸O-PIP$_2$/peak area of 17:0 20:4 PI(4,5)P$_2$ (internal standard) normalised to organic phosphate value as mean ± S.E.M. of *da/+; dPIP4K²⁹* (green) or *da> Mtm^WT::GFP; dPIP4K²⁹* (magenta). Biological samples n = 3, where each sample was made from five third instar wandering larvae. Unpaired *t* test with Welch's correction showed *P*-value = 0.7830 between *da/+; dPIP4K²⁹* and *da> Mtm^WT::GFP; dPIP4K²⁹*.

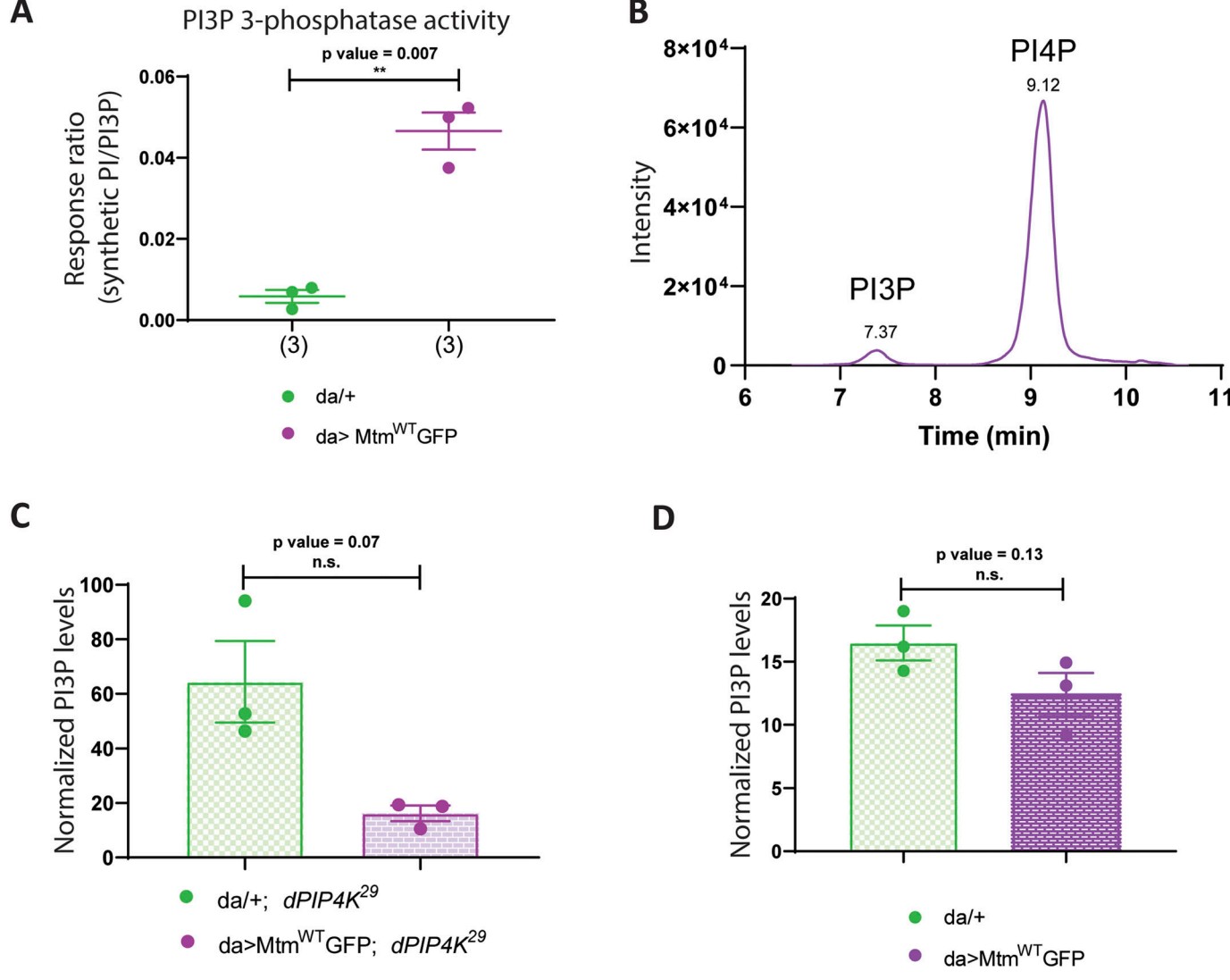

**Figure 3. Mtm reduces PI3P levels when overexpressed in *dPIP4K^29*.**
**(A)** In vitro phosphatase assay on synthetic PI3P. Graph representing the response ratio of 17:0 20:4 PI (product)/17:0 20:4 PI3P (substrate) formed as mean ± S.E.M. on addition of either control (*da/+*) or *da>Mtm^WT::GFP* lysates. Lysate samples = 3, where each sample was made from five third instar wandering larvae. unpaired *t* test with Welch correction showed *P*-value = 0.007. **(B)** Extracted ion chromatogram (XIC) of deacylated PI3P or GroPI3P (glycerophosphoinositol 3-phosphate) peak at $R_t$ = 7.37 min, separated from deacylated PI4P or GroPI4P (glycerophosphoinositol 4-phosphate) peak at $R_t$ = 9.12 min obtained from injecting WT larval lipid extract (details of sample preparation is discussed in methods). **(C)** Graph representing Normalised PI3P levels which is the peak area of GroPI3P/peak area of GroPI4P normalised to organic phosphate value of total lipid extracts as mean ± S.E.M. of *da/+; dPIP4K^29* (green) and *da> Mtm^WT::GFP; dPIP4K^29* (magenta). Biological samples = 3, where each sample was made from three third instar wandering larvae. unpaired *t* test with Welch correction showed *P*-value = 0.07. **(D)** Graph representing normalised PI3P levels which is the peak area of GroPI3P/peak area of GroPI4P normalised to organic phosphate value of total lipid extracts as mean ± S.E.M. of *da/+* (green) and *da> Mtm^WT::GFP* (magenta). Biological samples = 3, where each sample was made from three third instar wandering larvae. Unpaired *t* test with Welch correction showed *P*-value = 0.13.

devised an alternate assay to measure PI3P from larvae. Briefly, we developed an in vitro lipid kinase reaction using purified mCherry:: dFab1 to quantify PI3P from larval lipid extracts using radionuclide labelling (schematic in Fig S3A). Fig S3B(i) indicates the PI(3,5)P$_2$ spot on a TLC formed from PI3P during the in vitro kinase reaction. Although lipid extracts from WT and *dPIP4K^29* larvae showed similar PI(3,5)P$_2$ spot intensities on the TLC (Fig S3B(i)), normalisation of the PI(3,5)P$_2$ spot intensity against total organic phosphate levels in each sample confirmed that the total PI3P levels were higher in *dPIP4K^29* (Fig S3B(ii)) compared with controls. To confirm that the

increase of PI3P in *dPIP4K^29* was a result of the absence of PIP4K, we reconstituted WT *dPIP4K* in *dPIP4K^29* and measured PI3P (*Act5C> dPIP4K^WT::GFP; dPIP4K^29*) and observed that the elevated PI3P in *dPIP4K^29* was reverted to normal, indicating that dPIP4K can indeed regulate PI3P levels in vivo (Fig 4B). The catalytic activity of dPIP4K is essential to maintain salivary gland cell size (Mathre et al, 2019). Therefore, to check whether this catalytic activity was also necessary to control PI3P levels in vivo, we reconstituted *dPIP4K^29* with a catalytically inactive dPIP4K (*dPIP4K^D271A*) and measured PI3P (*Act5C> dPIP4K^D271A; dPIP4K^29*); we found that expressing catalytically

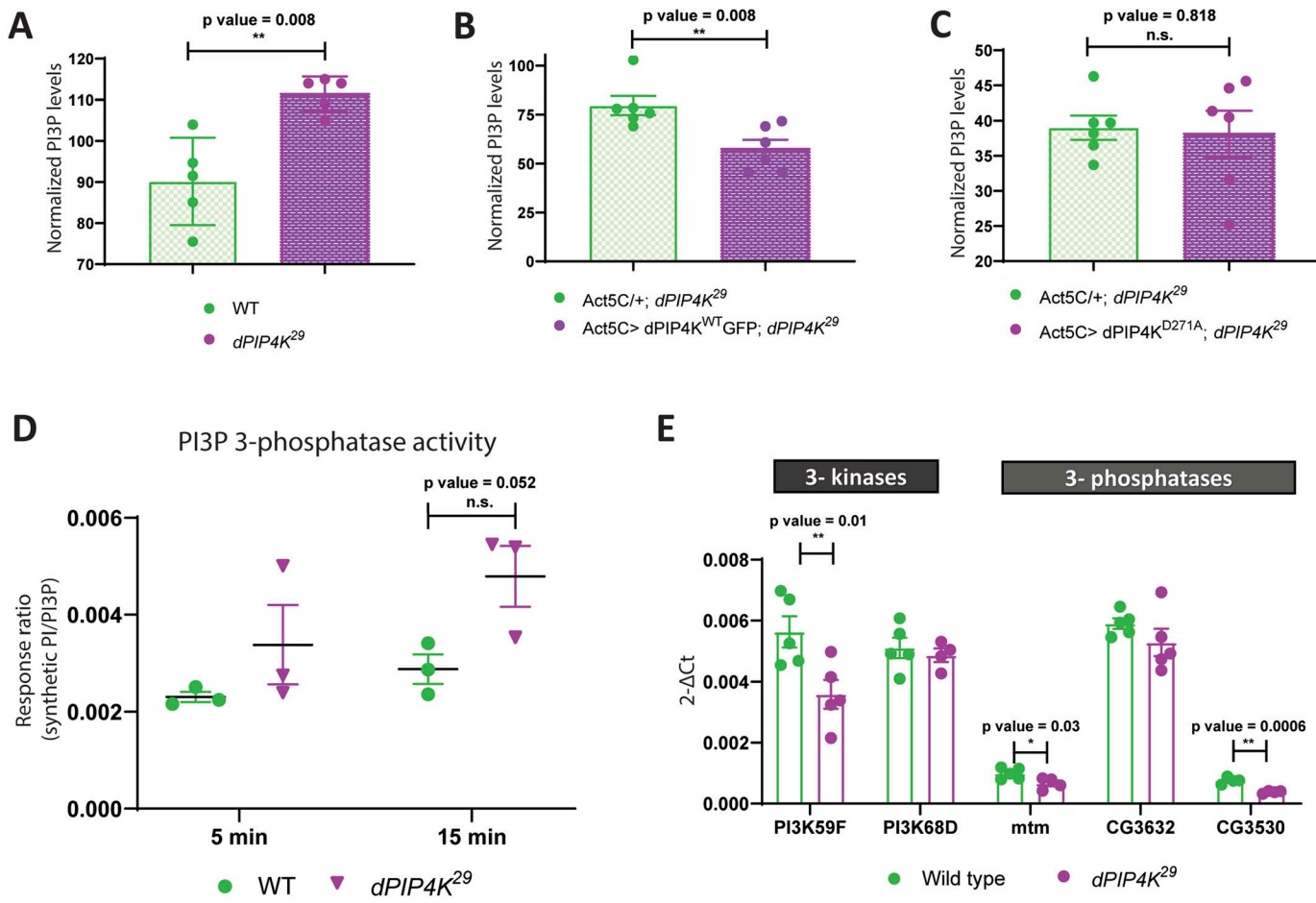

**Figure 4. dPIP4K regulates PI3P levels in vivo.**
**(A)** Graph representing Normalised PI3P levels which is the peak area of GroPI3P/peak area of GroPI4P normalised to organic phosphate value as mean ± S.E.M. of WT (green) and *dPIP4K^29* (magenta). Biological samples = 5, where each sample was made from five third instar wandering larvae. Unpaired *t* test with Welch correction showed *P*-value = 0.008. **(B)** Graph representing normalised PI3P levels which is the peak area of GroPI3P/peak area of GroPI4P normalised to organic phosphate value of total lipid extracts as mean ± S.E.M. of *Act5C/+; dPIP4K^29* (green) or *Act5C> dPIP4K^WT::GFP; dPIP4K^29* (magenta). Biological samples = 5, where each sample was made from three third instar wandering larvae. unpaired *t* test with Welch correction showed *P*-value = 0.008. **(C)** Graph representing normalised PI3P levels which is the peak area of GroPI3P/peak area of GroPI4P normalised to organic phosphate value of total lipid extracts as mean ± S.E.M. of *Act5C/+; dPIP4K^29* (green) or *Act5C> dPIP4K^D271A; dPIP4K^29* (magenta). Biological samples = 6, where each sample was made from three third instar wandering larvae. Unpaired *t* test with Welch correction showed *P*-value = 0.818. **(D)** In vitro phosphatase assay on synthetic PI3P. Graph representing the response ratio of 17:0 20:4 PI (product)/17:0 20:4 PI3P (substrate) formed as mean ± S.E.M. on addition of either WT or *dPIP4K^29* lysates for either a 5-min or a 15-min reaction. Lysate samples = 3, where each sample was made from five third instar wandering larvae. Multiple unpaired *t* test showed *P*-value = 0.26 for 5 min time point and *P*-value = 0.052. **(E)** qPCR measurements for mRNA levels of *PI3K59F* and *PI3K68D* from either WT (green) or *dPIP4K^29* (magenta). unpaired *t* test showed *P*-value = 0.01 for *PI3K59F* and *P*-value = 0.58 for *PI3K68D*. qPCR measurements for mRNA levels of *Mtm, CG3632,* and *CG3530* from either WT (green) or *dPIP4K^29* (magenta). Unpaired *t* test showed *P*-value = 0.03 for *Mtm*, *P*-value = 0.23 for *CG3632*, and *P*-value = 0.0006 for *CG3530*.

dead dPIP4K^D271A could not significantly decrease the levels of PI3P in *dPIP4K^29* (Fig 4C). These findings indicate that the catalytic activity of dPIP4K is required to regulate PI3P levels in vivo.

### Regulation of PI3P by dPIP4K is unlikely to be via indirect mechanisms

Although PI5P is the canonical in vivo substrate for dPIP4K, the enzyme can also use PI3P as a substrate with low efficiency (Gupta et al, 2013; Ghosh et al, 2019), a feature conserved with mammalian PIP4Ks (Zhang et al, 1997). In the context of our observation that PI3P levels are elevated in *dPIP4K^29*, dPIP4K could regulate PI3P levels either through its ability to directly phosphorylate this lipid or indirectly via its ability to regulate other enzymes that are established regulators of PI3P

levels (e.g., through negative regulation of PI 3-kinase activity or through positive regulation of a 3-phosphatase that dephosphorylate PI3P [Fig S3C]). A reduction in 3-phosphatase activity on PI3P in *dPIP4K^29* could lead to accumulation of PI3P. To test this possibility, the total 3-phosphatase activity of *dPIP4K^29* lysates was assessed. We did not observe a reduction in 3-phosphatase activity that might explain the elevated PI3P levels. In fact, there was an increase in response ratio (indicative of the 3-phosphatase activity on PI3P) in a 15-min in vitro assay in mutant lysates as compared with WT lysates (Fig 4D). However, we observed that the transcript levels of *Mtm* and *CG3530* —a putative 3-phosphatase were decreased in *dPIP4K^29* as compared with controls (Fig 4E), which does not directly correlate with the lack of decrease in total 3-phosphatase activity observed in Fig 4D. PI3K59F activity could not be directly measured from larval lysates; however,

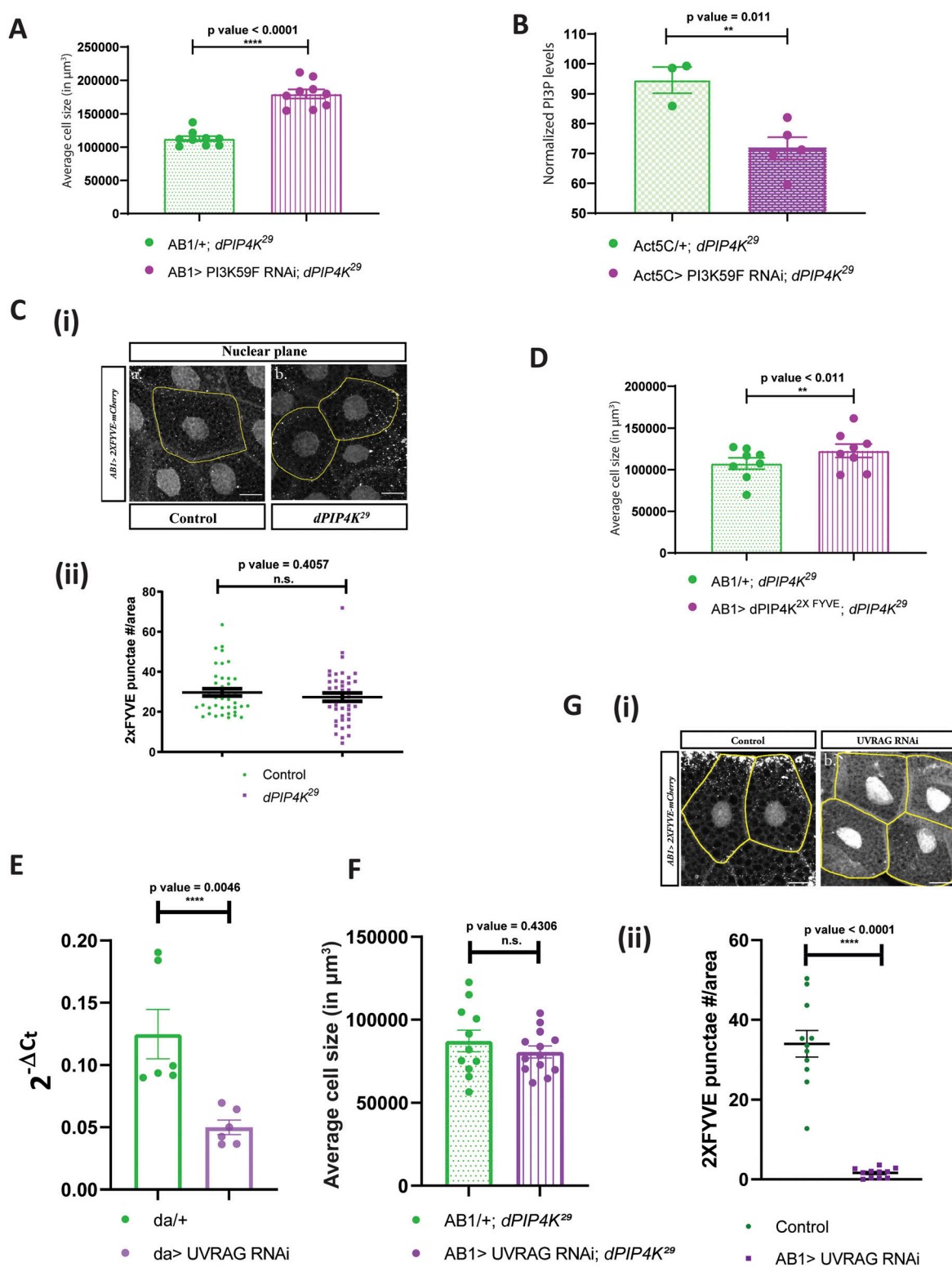

**Figure 5. PIP4K regulates a non-endosomal PI3P pool in *Drosophila* salivary glands.**
**(A)** Graph representing average cell size measurement (in µm³) as mean ± S.E.M. of salivary glands from wandering third instar larvae of *AB1/+; dPIP4K²⁹* (n = 9), *AB1>PI3K59F RNAi; dPIP4K²⁹* (n = 9). Sample size is represented on individual bars. Unpaired *t* test with Welch correction showed *P*-value < 0.0001. **(B)** Graph representing normalised PI3P levels which is the peak area of GroPI3P/peak area of GroPI4P normalised to organic phosphate value of total lipid extracts as mean ± S.E.M. of *Act5C/+; dPIP4K²⁹* (green) or *Act5C> PI3K59F RNAi; dPIP4K²⁹* (magenta). Biological samples = 5, where each sample was made from five third instar wandering larvae. Unpaired *t* test with Welch correction showed *P*-value = 0.011. **(C)** (i) Representative confocal *z*-projections depicting a subpopulation of early endosomal compartment using 2XFYVE-

we measured the mRNA expression of the two known PI 3-kinase genes—*PI3K59F* and *PI3K68D*. Although the transcript levels of *PI3K68D* were unchanged between *dPIP4K$^{29}$* and controls, we observed that *PI3K59F* transcripts were in fact lower in *dPIP4K$^{29}$* compared with controls (Fig 4E). Thus, it seems unlikely that up-regulation of the aforementioned PI 3-kinases or down-regulation of the 3-phosphatases contributes to the increased PI3P levels in *dPIP4K$^{29}$*. These findings led us to conclude that the regulation of PI3P levels by dPIP4K is unlikely via indirect mechanisms.

## PIP4K regulates cell size through a non-early endosomal PI3P pool

The major source of PI3P generation in cells is the class III PI3-kinase called Vps34, whose *Drosophila* ortholog is PI3K59F. Consequently, if PI3P generated by PI3K59F is relevant in regulating cell size in our model, we would predict a rescue of cell size by down-regulating PI3K59F in *dPIP4K$^{29}$*. We down-regulated PI3K59F activity using RNA interference (Fig S4A depicts the extent of *PI3K59F* transcript depletion) in *dPIP4K$^{29}$* background and indeed observed a rescue of the reduced cell size (Fig 5A); knockdown of *PI3K59F* in an otherwise WT background did not change cell size (Fig S4B). Furthermore, measurement of PI3P levels from *dPIP4K$^{29}$* larvae expressing PI3K59F RNAi showed a reduction in PI3P levels compared with *dPIP4K$^{29}$* (Fig 5B).

Broadly, PI3K59F is known to be functional at two locations in cells—namely the early endosomal compartment and at multiple steps of the autophagy pathway (Nascimbeni et al, 2017). To test if the early endosomal PI3P pool contributes to the reduced cell size in *dPIP4K$^{29}$*, we imaged the tandem FYVE domain fused to mCherry (2XFYVE::mCherry), a reporter for endosomal PI3P, in salivary glands of *dPIP4K$^{29}$* and compared it with WT. The 2XFYVE::mCherry probe revealed punctate structures which were perinuclear (Fig 5C(i)). Quantification of the number of punctae per unit area calculated for the perinuclear punctae showed no difference between WT and *dPIP4K$^{29}$* (Fig 5C(ii)) although the probe was expressed at equal levels in both genotypes (Fig S4C).

To further validate if a change of PI3P at the early endosomal location in *dPIP4K$^{29}$* was required to support cell size, we tagged dPIP4K with the tandem FYVE domain at the C-terminus end of the protein (dPIP4K$^{2XFYVE}$) to target it to the PI3P-enriched endosomal compartment (Kamalesh et al, 2017; Sharma et al, 2019) and reconstituted this in the background of *dPIP4K$^{29}$*. We did not observe a rescue in the cell size of *dPIP4K$^{29}$* under these conditions suggesting

that dPIP4K function at early endosomes is not sufficient for cell size regulation (Fig 5D).

Previous work in mammalian cells and yeast has shown that UVRAG associates with Vps34 to form PI3P that regulates endocytic trafficking (Kihara et al, 2001). We depleted UVRAG in *dPIP4K$^{29}$* (Fig 5E) and found that there was no effect on the cell size phenotype of *dPIP4K$^{29}$* (Fig 5F), although the depletion of UVRAG under these conditions was sufficient to reduce endosomal PI3P as measured by the 2XFYVE::mCherry probe (Fig 5G(i and ii)). Under these conditions, UVRAG depletion was unable to reduce Atg8a punctae in *dPIP4K$^{29}$* (data not shown).

## PIP4K regulates autophagy and lysosomal function

The other subcellular location at which a Vps34-regulated PI3P pool is important is the early autophagosomal membranes. We were unable to directly measure the PI3P pool at autophagosomal membranes. However, an increase in PI3P levels at this compartment would lead to an increase in the extent of autophagy (Burman & Ktistakis, 2010) and can be assayed by an increase in the *Drosophila* ortholog of microtubule-associated protein 1A/1B-light chain 3 (LC3) called Atg8a. We expressed Atg8a::mCherry in salivary glands and the probe was expressed at equal levels in both genotypes (Fig S4D). Measurement of the number of Atg8a::mCherry punctae showed these were increased in *dPIP4K$^{29}$* glands compared with controls (Fig 6A(i and ii)).

PI3P in the early autophagy pathway is generated by Vps34 and regulated by Atg1-mediated activation of the Vps34 Complex I components Beclin-1 and Atg14, after which, early autophagic membranes mature into autophagosomes (King et al, 2021). It has been demonstrated that induction of autophagy by overexpressing Atg1 can cause a decrease in cell size of fat body cells in *Drosophila* larvae (Scott et al, 2007). Likewise, in this study, we found that the overexpression of Atg1 in the salivary glands of *Drosophila* larvae caused a drastic decrease in cell size (Fig S4E). Importantly, down-regulating Atg1 activity in *dPIP4K$^{29}$* could reverse the cell size phenotype in salivary glands (Fig 6B), whereas no change was observed in otherwise WT background (Fig S4F), indicating that the autophagy pathway is up-regulated in *dPIP4K$^{29}$*. We reasoned that if the elevated PI3P in *dPIP4K$^{29}$* causes an up-regulation in autophagy leading to cell size reduction, then by down-regulating Atg8a, that functions downstream of PI3P mediated induction of autophagy, we would be able to reverse the phenotype of cell size decrease. We indeed observed that down-regulation of Atg8a in *dPIP4K$^{29}$* caused a reversal of the reduced cell size (Fig 6C), whereas there was no significant change in cell size by down-regulation of Atg8a in an otherwise WT

mCherry in the salivary glands from wandering third instar larvae of *AB1> 2XFYVE::mCherry* (Control) and *AB1 2XFYVE::mCherry; dPIP4K$^{29}$*. Scale bar indicated at 20 μm. (ii) Graph representing 2XFYVE punctae measurement in the salivary glands from wandering third instar larvae of *AB1> 2XFYVE::mCherry* (N = 8, n = 40) and *AB1>2XFYVE::mCherry; dPIP4K$^{29}$* (N = 8, n = 40). Unpaired *t* test with Welch correction showed *P*-value = 0.4057. **(D)** Graph representing average cell size measurement (in μm$^3$) as mean ± S.E.M. of salivary glands from wandering third instar larvae of *AB1/+; dPIP4K$^{29}$* (n = 8), *AB1>dPIP4K$^{2XFYVE}$; dPIP4K$^{29}$* (n = 8). Sample size is represented on individual bars. Unpaired *t* test with Welch correction showed *P*-value = 0.171. **(E)** qPCR measurements for mRNA levels of *UVRAG* from either *da/+* (green) or *da> UVRAG* RNAi (magenta). Unpaired *t* test showed *P*-value = 0.0046. (n = 6, where each n is derived from five third instar wandering larvae). **(F)** Graph representing average cell size measurement as mean ± S.E.M. of salivary glands from wandering third instar larvae of *AB1/+; dPIP4K$^{29}$* (n = 12) and *AB1> UVRAG* RNAi; *dPIP4K$^{29}$* (n = 12). unpaired *t* test with Welch correction showed *P*-value = 0.4306. **(G)** (i) Representative confocal z-projections depicting 2XFYVE punctae using 2XFYVE::mCherry in the salivary glands from the genotypes a. *AB1 2XFYVE::mCherry* (Control) b. *AB1 2XFYVE::mCherry; UVRAG* RNAi. (ii) Graph representing measurement of 2XFYVE punctae numbers per unit cell area of the salivary glands from wandering third instar larvae of *AB1> 2XFYVE::mCherry* (N = 3, n = 11), *AB1> 2XFYVE::mCherry; UVRAG* RNAi (N = 3,n = 11).

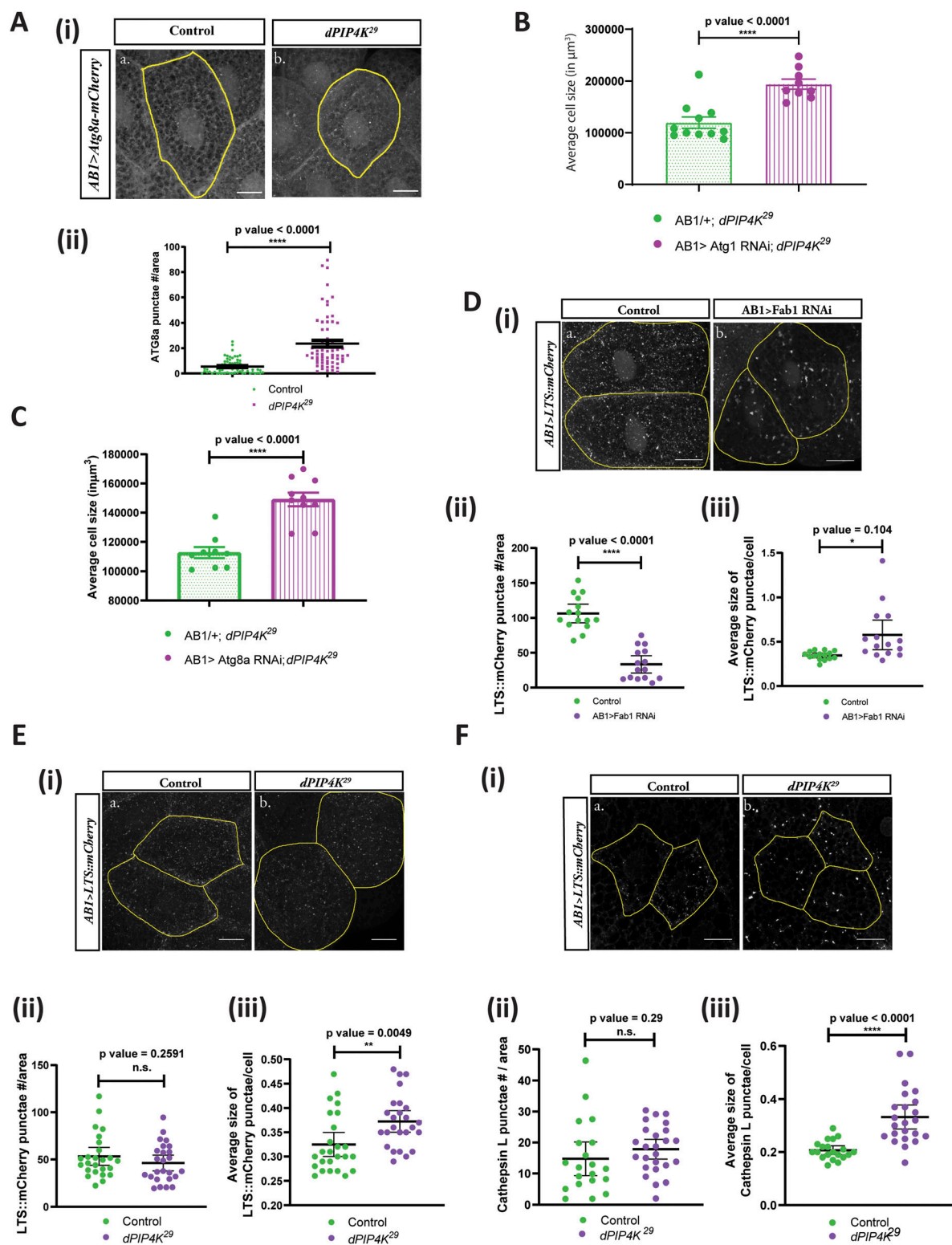

**Figure 6. PIP4K affects the auto-lysosomal pathway to affect cell size in *Drosophila* salivary glands.**
**(A)** (i) Representative confocal z-projections depicting autophagosomal levels using Atg8a-mCherry in the salivary glands from the wandering third instar larvae of *AB1>Atg8a::mCherry* (Control) and *AB1>Atg8a::mCherry; dPIP4K²⁹*. Scale bar indicated at 20 μm. (ii) Graph representing Atg8a punctae measurement in the salivary glands from wandering third instar larvae of *AB1>Atg8a::mCherry* (N = 10, n = 60) and *AB1>Atg8a::mCherry; dPIP4K²⁹* (N = 10, n = 62). Unpaired *t* test with Welch correction showed *P*-value < 0.0001. **(B)** Graph representing average cell size measurement (in μm³) as mean ± S.E.M. of salivary glands from wandering third instar larvae of *AB1/+; dPIP4K²⁹* (n = 11), *AB1>Atg1* RNAi; *dPIP4K²⁹* (n = 9). Sample size is represented on individual bars. Unpaired *t* test with Welch correction showed *P*-value < 0.0001. **(C)** Graph

background (Fig S4G). Importantly, down-regulation of Atg8a using the same RNAi line was able to decrease the number of Atg8a::mCherry in salivary glands (Fig S4H(i and ii)); the expression of the probe being equivalent between both genotypes (Fig S4I).

If autophagy is enhanced in $dPIP4K^{29}$, one might expect to see alterations in downstream compartments of the pathway such as the number or morphology of lysosomes that arise from fusion with autophagosomes. We tested this by measuring the number and size of lysosomes; to do this, we generated a transgenic reporter construct in which the lysosome-targeting sequence (LTS) is fused to mCherry. When expressed in salivary glands, this reporter marks lysosomes as punctate structures whose number, size, and distribution can be studied. Using this reporter, we studied the impact of Fab1 depletion on lysosomes in salivary glands (Fig 6D(i)). As previously published, we found that depletion of Fab1 results in a reduction in the number of lysosomes (Fig 6D(ii)) and the average size of lysosomal punctae was increased (Fig 6D(iii)). In $dPIP4K^{29}$ (Fig 6E(i)), we found that there was no reduction in the number of LTS:: mCherry punctae (Fig 6E(ii)) but the average size of punctae was increased (Fig 6E(iii)). Western blotting of the probe level from salivary glands of WT and $dPIP4K^{29}$ larvae did not reveal any significant difference (Fig S5A). We also stained salivary glands with an antibody to Cathepsin L (Fig 6F(i)), a lysosomal protease that marks active lysosomes and also found that whereas the number of active lysosomes was not increased in $dPIP4K^{29}$ (Fig 6F(ii)), the average size of lysosomal punctae was increased (Fig 6F(iii)).

### PI3P regulates cell size in salivary glands

We tested the effect of modulating PI3P levels in otherwise WT salivary glands. Depletion of *Mtm*, using an RNAi line has previously shown to increase PI3P levels in *Drosophila* (Velichkova et al, 2010; Jean et al, 2012). Using qPCR analysis, we validated that the RNAi reagent causes specific down-regulation of *Mtm* transcripts in *Drosophila* (Fig 7A). Interestingly, expressing Mtm RNAi in salivary glands caused a significant decrease in cell size (Fig 7B(i and ii)). Myotubularins are known to dimerize in cells and Mtm harbours a C-terminal coiled-coil domain which can potentially aid in dimerization (Jean et al, 2012). We observed a similar but smaller reduction in cell size when a catalytically dead version of Mtm ($Mtm^{D402A}$) that is expected to act as dominant negative construct was expressed in salivary glands (Fig S5B). Furthermore, measurement

of PI3P levels revealed a modest up-regulation of PI3P levels when measured from larvae expressing the Mtm RNAi (*da> Mtm RNAi*, Fig 7C). Therefore, Mtm downregulation in an otherwise WT background can cause PI3P elevation and cell size decrease in *Drosophila*.

To understand if the reduction in cell size brought about by down-regulating Mtm in salivary glands causes an up-regulation of the autophagic pathway much like in $dPIP4K^{29}$ mutants, we measured the number of Atg8a::mCherry in salivary glands of Mtm RNAi (Fig 7D(i)). It was observed that there was a substantial increase in the number of Atg8a::mCherry punctae as quantified in Fig 7D(ii), although the expression of the probe was equivalent in both genotypes (Fig S5C). Interestingly, depleting Atg8a in salivary glands also depleted of Mtm could partially rescue cell size as compared with glands where Mtm alone was down-regulated (Fig 7E). These findings corroborate the relationship of increased PI3P levels to the up-regulation of autophagy which can eventually contribute to a decrease in cell size of salivary glands of *Drosophila* larvae.

## Discussion

Conceptually, the cellular function of any enzyme can arise from its ability to regulate the levels of either its substrate or the product formed. When PIP4K was originally described (Rameh et al, 1997), its ability to generate the product $PI(4,5)P_2$ was recognised. However, $PI(4,5)P_2$, can also be synthesized from phosphatidylinositol 4 phosphate (PI4P) by the activity of phosphatidylinositol 4 phosphate 5 kinase (PIP5K) (reviewed in Kolay et al [2016]). Because PI4P is ca.10 times more abundant than PI5P, loss of PIP4K activity is unlikely to impact the overall levels of cellular $PI(4,5)P_2$. Consistent with this, knockout of *PIP4K* does not reduce the overall level of $PI(4,5)P_2$ ([Gupta et al, 2013] discussed in Kolay et al [2016]). In this study, a switch mutant version of PIP4K that can generate $PI(4,5)P_2$ from PI4P but not PI5P was unable to rescue the reduced cell size in $dPIP4K^{29}$ implying that the biochemical basis of dPIP4K function in supporting cell size is not by its product $PI(4,5)P_2$. Rather, given that the kinase activity of the enzyme is required for normal salivary gland cell size (Mathre et al, 2019), our findings imply that the regulation of the substrate levels by dPIP4K is likely to be relevant to the control of cell size. It was vital to be able to measure the levels of putative substrates of PIP4K from *Drosophila*

representing average cell size measurement (in µm3) as mean ± S.E.M. of salivary glands from wandering third instar larvae of *AB1/+; dPIP4K$^{29}$* (n = 9), *AB1>Atg8a* RNAi; *dPIP4K$^{29}$* (n = 9). Sample size is represented on individual bars. Unpaired *t* test with Welch correction showed *P*-value < 0.0001. **(D)** (i) Representative confocal z-projections depicting lysosomes using LTS::mCherry in the salivary glands from the genotypes a. *AB1>LTS::mCherry* (control) and b. *AB1>LTS::mCherry; Fab1* RNAi. Scale bar indicated at 20 µm. (ii) Graph representing measurement of LTS::mCherry punctae numbers per unit cell area of the salivary glands from wandering third instar larvae of *AB1>LTS:: mCherry* (N = 3,n = 15) and *AB1>LTS::mCherry; Fab1*RNAi (N = 3,n = 15). Unpaired *t* test with Welch correction showed *P*-value < 0.0001. (iii) Graph representing measurement of average size of LTS::mCherry punctae per unit cell of the salivary glands from wandering third instar larvae of *AB1>LTS::mCherry* (N = 3,n = 15) and *AB1>LTS::mCherry; Fab1*RNAi (N = 3,n = 15). Unpaired *t* test with Welch correction showed *P*-value = 0.0104. **(E)** (i) Representative confocal z-projections depicting lysosomes using LTS:: mCherry in the salivary glands from the genotypes a. *AB1>LTS::mCherry* (Control) and b. *AB1>LTS::mCherry; dPIP4K$^{29}$*. Scale bar indicated at 20 µm (ii) Graph representing measurement of LTS::mCherry punctae numbers per unit cell area of the salivary glands from wandering third instar larvae of *AB1>LTS::mCherry* (N = 5,n = 25) and *AB1>LTS:: mCherry; dPIP4K$^{29}$* (N = 2,n = 25). Unpaired *t* test with Welch correction showed *P*-value = 0.2591. (iii) Graph representing measurement of average size of LTS::mCherry punctae per unit cell of the salivary glands from wandering third instar larvae of *AB1>LTS::mCherry* (N = 5,n = 25) and *AB1>LTS::mCherry; dPIP4K$^{29}$* (N = 5,n = 25). Unpaired *t* test with Welch correction showed *P*-value = 0.0049. **(F)** (i) Representative confocal z-projections depicting lysosomes using Cathepsin L staining in the salivary glands from the genotypes a. *w$^{1118}$* (control) and b. *dPIP4K$^{29}$* (mutant). Scale bar indicated at 20 µm (ii) Graph representing measurement of Cathepsin L punctae numbers per unit cell area of the salivary glands from wandering third instar larvae of *w$^{1118}$* (control) and *dPIP4K$^{29}$* (mutant). unpaired *t* test with Welch correction showed *P*-value = 0.2900. (iii) Graph representing measurement of average size of Cathepsin L punctae per unit cell of the salivary glands from wandering third instar larvae of *w$^{1118}$* (control) and *dPIP4K$^{29}$* (mutant). Unpaired *t* test with Welch correction showed *P*-value < 0.0001.

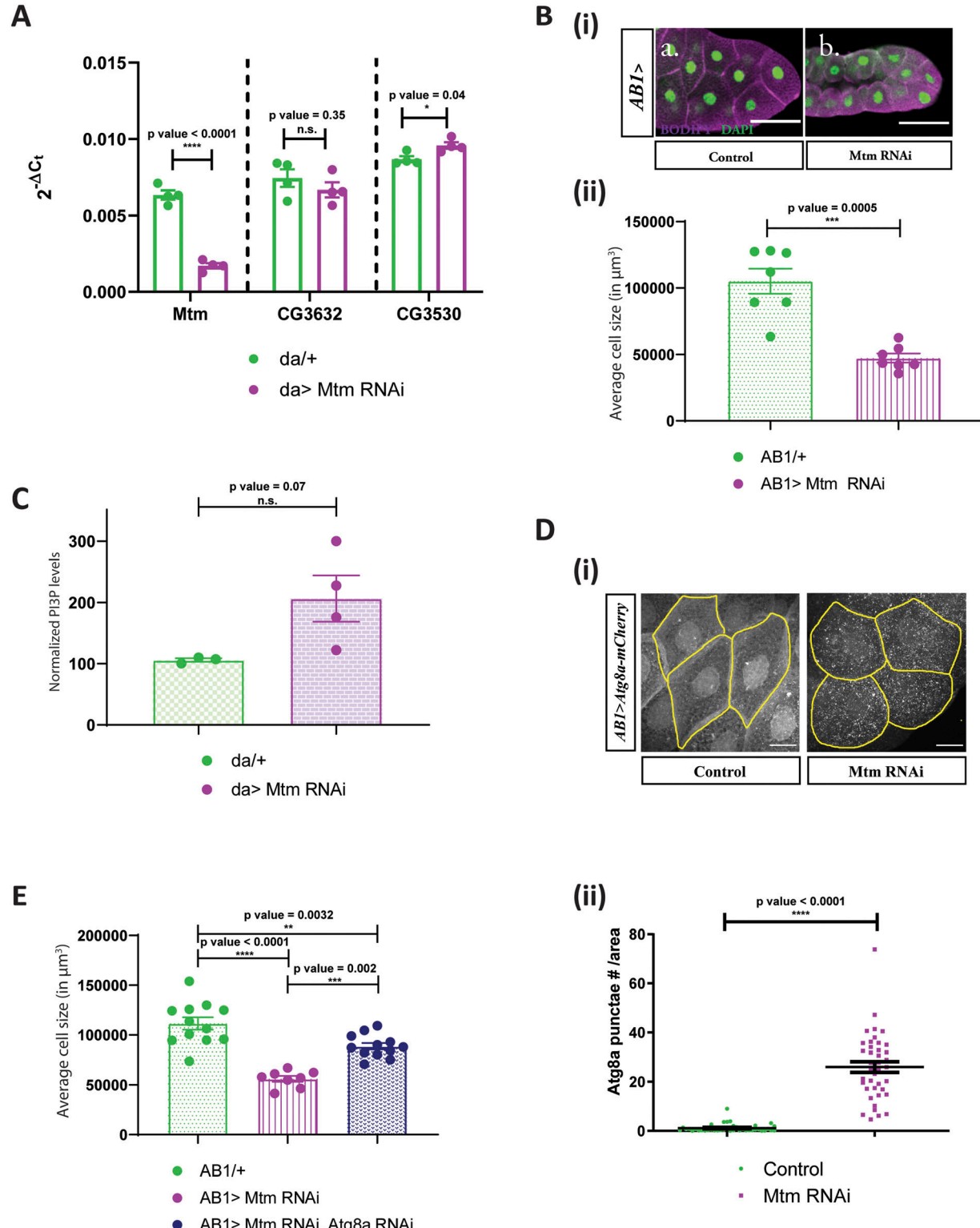

**Figure 7. PI3P regulates cell size in salivary glands independent of dPIP4K.**

**(A)** qPCR measurements for mRNA levels of *Mtm, CG3632, and CG3530* from either control (*da/+*, in green) or *da> Mtm* RNAi, in magenta. Multiple *t* test with post hoc Holm-Sidak's test showed *P*-value < 0.0001 between *da/+* and *da> Mtm* RNAi for *Mtm* and *P*-value = 0.35 between *da/+* and *da> Mtm* RNAi for *CG3632*, and *P*-value = 0.04 between *da/+* and *da> Mtm* RNAi for *CG3530*. **(B)** (i) Representative confocal images of salivary glands from the genotypes a. *AB1/+*, b. *AB1>Mtm* RNAi. Cell body is marked magenta by BODIPY-conjugated lipid dye, and the nucleus is marked by DAPI shown in green. Scale bar indicated at 50 μm. (ii) Graph representing average cell size measurement (in μm³) as mean ± S.E.M. of salivary glands from wandering third instar larvae of *AB1/+* (n = 7), *AB1> Mtm* RNAi (n = 7). Sample size is represented on individual bars. Unpaired *t*

tissues and therefore, we developed a label-free LC-MS/MS-based method to detect and quantify PI3P and PI5P. Previously, researchers used more cumbersome radioactivity-based detection to measure PI3P and PI5P (Chicanne et al, 2012; Jones et al, 2013); however, new label-free methods are being reported to quantify PIPs (Ghosh et al, 2019; Morioka et al, 2022). In this study, we report the use of a label-free LC-MS/MS-based method to measure PI3P levels in vivo from *Drosophila* tissues. In future, this method can also be used to measure PI3P levels from tissues of other model organisms to address key questions in PI3P biology.

Given that previous studies have identified PI5P as the substrate best utilised by PIP4K and that PI5P levels are elevated in *dPIP4K[29]*, we expected that the reduced cell size phenotype in this mutant will be mediated by PI5P levels. However, in the course of this study, we found that (i) the expression of Mtm, a 3-phosphatase that is able to generate PI5P in vitro from PI(3,5)P$_2$ rescued the reduced cell size in *dPIP4K[29]*, (ii) the rescue of cell size in *dPIP4K[29]* by Mtm over-expression was not associated with a reduction in the levels of PI5P. These observations are not consistent with a role for elevated PI5P levels in the reduced cell size phenotype of *dPIP4K[29]*. Because PI3P has also been shown to be a substrate of dPIP4K in vitro, albeit with less efficiency (Gupta et al, 2013; Ghosh et al, 2019), we investigated PI3P levels in fly tissues and found that (i) PI3P levels were elevated in *dPIP4K[29]*, (ii) were rescued by reconstitution with a WT dPIP4K transgene but not a kinase dead version, (iii) the rescue of cell size in *dPIP4K[29]* by expression of Mtm was associated with a reduction in the elevated PI3P levels. Together, these findings strongly suggest that the elevated PI3P levels in *dPIP4K[29]* underpin the reduced salivary gland cell size. We also observed that depletion of Mtm in otherwise WT cells results in elevated PI3P levels and reduced cell size. This observation defines a role for PI3P as a regulator of cell size in *Drosophila*. In mammals, the major route of synthesis of PI3P is through the action of a class III PI3-kinase called Vps34. Down-regulating PI3K59F, the ortholog of Vps34 in *Drosophila*, reversed PI3P levels and cell size of *dPIP4K[29]* salivary glands. These findings define PI3P as a regulator of cell size in *Drosophila* salivary glands with Vps34 and dPIP4K working together to regulate PI3P levels in this setting. Interestingly, in a recent study of MEFs grown in culture, down-regulation of PIP4Kγ led to an increase in PI3P and PI(3,5)P$_2$ levels along with an expected rise in PI5P levels (Al-Ramahi et al, 2017). Another study in cultured mammalian epithelial cells has indicated that PI3P generated by Class II PI3K may regulate cell size in the context of shear stress over defined periods of time (Boukhalfa et al, 2020).

Our findings that demonstrate a key role for the kinase activity of dPIP4K in the regulation of PI3P levels in vivo raise the question of how the enzyme might perform this function. One possibility is that the kinase activity of dPIP4K regulates other lipid kinases and phosphatases whose activity regulates PI3P levels. However, in this study we did not find evidence to support this model. Another alternative is that the kinase activity of dPIP4K on PI3P might directly regulate its levels. Compared with PI5P, PIP4Kα has a lower but significant in vitro kinase activity towards PI3P as substrate (Zhang et al, 1997; Ghosh et al, 2019) and purified *Drosophila* PIP4K also shows modest PI(3,4)P$_2$ generation measured using a radioactive kinase assay indicating its 4-kinase activity on PI3P in vitro (Gupta et al, 2013). However, because of the fold difference in the in vitro activity between the two substrates, a role for PIP4K in regulating PI3P levels in vivo has not been considered. The results presented in this study strongly indicate that under optimal conditions in vivo, when PIP4K can access a pool of PI3P, the enzyme could metabolise PI3P. Future experiments to monitor the formation of the product of dPIP4K activity on PI3P, that is, PI(3,4)P$_2$ in vivo will help define the extent to which this reaction occurs in the context of intact cellular membranes.

What is the mechanism by which cell size is reduced in *dPIP4K[29]*? It has been noted in several studies of mammalian models that loss of PIP4K function is associated with an increase in either the initiation step or flux of autophagy (Vicinanza et al, 2015; Al-Ramahi et al, 2017; Lundquist et al, 2018). TORC1 activity is reported to be down-regulated in *dPIP4K[29]* (Gupta et al, 2013). Consistent with the known inverse relationship of TORC1 activity and autophagy (Kim & Guan, 2015), we found that levels of autophagy in *dPIP4K[29]* salivary gland cells was increased and the reduced cell size could be rescued by depletion of Atg1 and Atg8a, two key mediators of autophagy (Fig 8). Thus, enhanced autophagy underlies the reduced cell size in *dPIP4K[29]* salivary gland cells.

In the context of studies of PIP4K, it has been reported in human cells that PI5P can initiate autophagy and can even take over the function of PI3P to initiate autophagy in wortmannin-treated cells (Vicinanza et al, 2015). Surprisingly, we found that the cell size phenotype of *dPIP4K[29]* could be rescued just by altering PI3P levels, without any change in PI5P levels. Reducing PI3P levels by genetic knockdown of Vps34 (Class III PI3K) rescued the reduced cell size phenotype in *dPIP4K[29]* implying a role for PI3P synthesized by Vps34 in regulating cell size. Vps34 has been reported to synthesize PI3P primarily at two cellular compartments, early endosomes and autophagosomes. Multiple lines of evidence present in this study show that the early endosomal pool of PI3P generated by Vps34 does not contribute to the cell size phenotype of *dPIP4K[29]*. However, we found that (i) inhibition of autophagy by depletion of Atg8a, a protein required downstream of PI3P formation at the phagophore membrane during autophagosome biogenesis, rescues the cell size phenotype of *dPIP4K[29]*; (ii) depletion of the 3′-phosphatase, Mtm results in elevated PI3P levels, enhanced autophagy, and reduced salivary gland cell size; (iii) overexpression of Mtm in *dPIP4K[29]* resulted in a rescue of the cell size phenotype of the single mutant. Taken together, these observations strongly suggest that the regulation of PI3P by dPIP4K is required for normal autophagy in salivary gland cells (Fig 8).

---

test with Welch correction showed *P*-value = 0.0005. **(C)** Graph representing normalised PI3P levels which is the peak area of GroPI3P/peak area of GroPI4P normalised to organic phosphate value as mean ± S.E.M. of da/+ (green) and *da> Mtm* RNAi (magenta). Biological samples = 4, where each sample was made from five third instar wandering larvae. Unpaired *t* test with Welch correction showed *P*-value = 0.07. **(D)** (i) Representative confocal z-projections depicting autophagosomal levels using Atg8a-mCherry in the salivary glands from the genotypes a. *AB1>Atg8a::mCherry* (control), b. *AB1>Atg8a::mCherry; Mtm* RNAi. Scale bar indicated at 20 μm. (ii) Graph representing Atg8a punctae measurement in the salivary glands from wandering third instar larvae of *AB1>Atg8a::mCherry* (N = 7, n = 40), *AB1>Atg8a::mCherry; Mtm* RNAi (N = 7, n = 40). Unpaired *t* test with Welch correction showed *P*-value < 0.0001. **(E)** Graph representing average cell size measurement (in μm$^3$) as mean ± S.E.M. of salivary glands from wandering third instar larvae of *AB1/+* (n = 11), *AB1>Mtm* RNAi (n = 8), *AB1>Mtm* RNAi, *Atg8a* RNAi (n = 12). Sample size is represented on individual bars. One-way ANOVA with post hoc Tukey's test showed *P*-value < 0.0001 between *AB1/+* and *AB1>Mtm* RNAi and *P*-value = 0.0002 between *AB1/+; dPIP4K[29]* and *AB1>Mtm* RNAi, *Atg8a* RNAi.

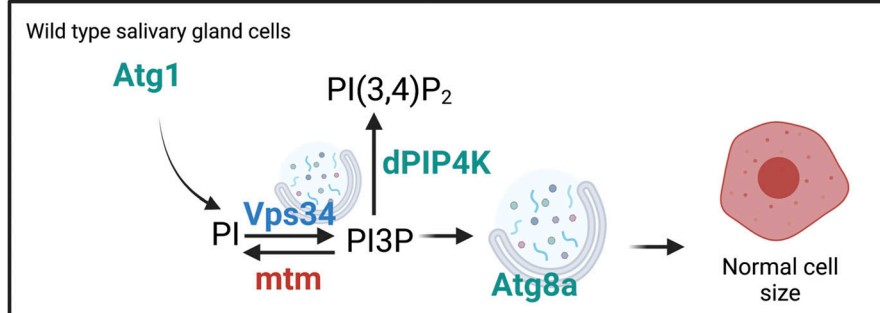

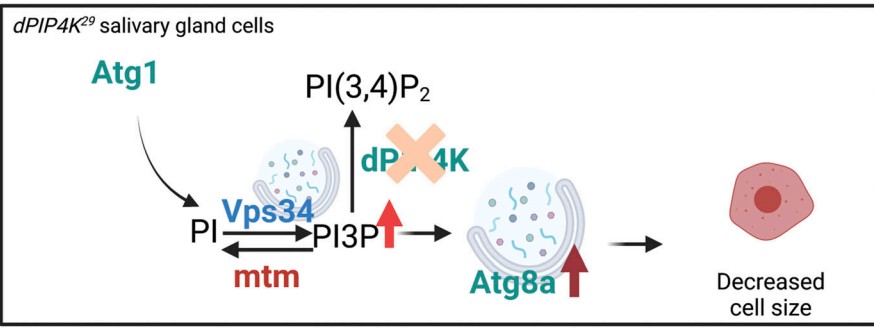

**Figure 8. Regulation of autophagy by PIP4K.** Schematic showing the regulation of PI3P at an autophagic membrane downstream of Atg1. In this study, dPIP4K which metabolizes PI3P has been shown to genetically interact with Vps34 and Mtm, two established PI3P regulators. Atg8a-marked autophagosomes form, mature to autolysosomes, and function in maintaining normal cell size in WT cells. In $dPIP4K^{29}$ cells, PI3P levels increase, leading to an increase in Atg8a punctae by increased autophagy initiation, which in turn leads to decreased cell size phenotype.

Within the autophagy pathway, PI3P levels can regulate the process at more than one location. A role for Vps34-dependent PI3P generation during the initiation of autophagy is well known and, in our system, PIP4K could control this pool of PI3P. However, in *Drosophila*, in contrast to the early endosomal system, to date, there in no probe that can be used to measure this specific pool of PI3P. The development of such a tool will allow a direct test of the model that dPIP4K regulates the PI3P pool that controls autophagy. A second location at which PI3P plays a critical role is the amphisome–lysosome fusion stage where PI3P is used to generate PI(3,5)P$_2$ by Fab1 (Rusten et al, 2007); this step is required for the fusion of endosome/autophagosomes to lysosomes. Interestingly, depletion of Fab1 in salivary gland results in a reduction in the number of lysosomes but an increase in their average size. In contrast, we found that in $dPIP4K^{29}$, the number of lysosomes was not reduced although the average size of individual lysosomes was enhanced. These observations may be indicative of an enhanced flux through the autophagosomal system although the present data cannot completely exclude a component of fusion defect. Although further analysis will be required to fully understand the mechanism by which PIP4K regulates autophagy, our data provide compelling evidence that the elevated levels of PI3P in $dPIP4K^{29}$ induces enhanced autophagy leading to reduction in cell size.

## Materials and Methods

### Fly culture strains and plasmid construction

All experiments were performed with *Drosophila melanogaster* (hereafter referred to as *Drosophila*). Cultures were reared on standard medium containing corn flour, sugar, yeast powder, and agar along with antibacterial and antifungal agents. Genetic crosses were set up with Gal4 background strains and maintained at 25°C and 50% relative humidity (Brand & Perrimon, 1993). There was no internal illumination within the incubator and the larvae of the correct genotype were selected at the third instar wandering stage using morphological criteria. *Drosophila* strains used were Oregon-R and $w^{1118}$ (WT strain), $dPIP4K^{29}$ (homozygous null mutant of dPIP4K made by Raghu lab), da-Gal4 (*da*), Act5C-Gal4/CyoGFP (*Act5C*), AB1-Gal4 (*AB1*), UAS hPIP4K2B/TM6Tb, UAS hPIP4K2B[A381E]/TM6Tb, Mtm[WT]::GFP (Amy Kiger, UCSD), Mtm-RNAi (#AK0246; Amy Kiger, UCSD), Mtm[D402A]GFP (Amy Kiger, UCSD) UAS dPIP4K[WT]::GFP (our lab), UAS dPIP4K[D271A](untagged) (our lab), UAS PI3K59F RNAi (v100296; VDRC), Atg1 RNAi (44034; Bloomington), Atg8a RNAi (34340; Bloomington), w; UAS-*mCherry::2XFYVE*[2] (Amy Kiger, UCSD), UAS-mCherry::Atg8a (37750; Bloomington), UVRAG RNAi (34368; Bloomington), Fab1 RNAi (27591; VDRC), UAS-LTS::mCherry (our lab). *dFab1* sequence was subcloned using Gibson assembly from clone GH27216 obtained from *Drosophila* Genomics Resource Center (DGRC) into pUAST-attB-mChherry vector to generate N-terminally tagged mCherry-dFab1. The 39-amino acid-long signal sequence from p18/LAMTOR known as the LTS was obtained from the construct lysosomal-dPIP4K::eGFP (Sharma et al, 2019). The LTS signal peptide was cloned into pUAST-attB-mCherry vector using Gibson assembly and was introduced upstream to the mCherry sequence. The construct was microinjected to obtain transgenic flies.

### S2R+ cells: culturing and transfection

*Drosophila* S2R+ cells were cultured and maintained as mentioned earlier (Gupta et al, 2013). Transient transfections for 48 h were performed as mentioned previously (Mathre et al, 2019). Primers for amplifying dsRNA template against *Drosophila* genes were selected from DRSC/TRiP Functional Genomics Resources after confirming

specificity of primers. A T7 RNA polymerase promoter sequence (5'-TAATACGACTCACTATAGGGAGA-3') was added at the 5' end of the primers for the T7 DNA-dependent RNA polymerase to bind during in vitro transcription. The dsRNA was synthesised using amplicons amplified from BDGP gold clones (Mtm: LD28822, CG3632: LD11744, and CG3530: GH04637), purchased from DGRC.

### RNA extraction and qPCR analysis

RNA was extracted from *Drosophila* S2R+ cells using TRIzol reagent (15596018; Life Technologies). Purified RNA was treated with amplification grade DNase I (18068015; Thermo Fisher Scientific). cDNA conversion was done using SuperScript II RNase H– Reverse Transcriptase (18064014; Thermo Fisher Scientific) and random hexamers (N8080127; Thermo Fisher Scientific). Quantitative PCR (qPCR) was performed using Power SybrGreen PCR master-mix (4367659; Applied Biosystems) in an Applied Biosystem 7500 Fast Real Time PCR instrument. Primers were designed at the exon–exon junctions following the parameters recommended for qPCR. Transcript levels of the ribosomal protein 49 (RP49) were used for normalization across samples. Three separate samples were collected from each treatment, and duplicate measures of each sample were conducted to ensure the consistency of the data.

### Western blotting and immunoprecipitation

#### *Western blots*
Salivary glands or larval samples were made exactly as mentioned in our previous work (Ghosh et al, 2019; Mathre et al, 2019). Dilutions of antibodies used: 1:4,000 for anti-tubulin (E7-c), (mouse) from

**Following are the list of primers used for the in vitro transcription of dsRNA.**

| Primer name | Sequence |
|---|---|
| Mtm dsRNA II F (DRSC36764) | 5'-TAATACGACTCACTATAGGGAGAACTCGTCGCTGGACCAGTAT-3' |
| Mtm dsRNA II R (DRSC36764) | 5'-TAATACGACTCACTATAGGGAGAATGCGTACAAGTAGGGGGAA-3' |
| CG3632 dsRNA II F (DRSC36821) | 5'-TAATACGACTCACTATAGGGAGAACCATCGAGAAGAATGGACG-3' |
| CG3632 dsRNA II R (DRSC36821) | 5'-TAATACGACTCACTATAGGGAGAATAGGAACGTGCCGAAGAGA-3' |
| CG3530 dsRNA I F (DRSC36794) | 5'-TAATACGACTCACTATAGGGAGAGCTCGATAGCAAGGAGCACT-3' |
| CG3530 dsRNA I R (DRSC36794) | 5'-TAATACGACTCACTATAGGGAGACAGGAGCAGGTGGTTACGTT-3' |
| GFP ds RNA F | 5'-TAATACGACTCACTATAGGGATGGTGAGCAAGGGCGAGGAG-3' |
| GFP ds RNA R | 5'-TAATACGACTCACTATAGGGCTTGTACAGCTCGTCCATGCCG-3' |

DSHB, 1:1,000 for anti-actin antibody (Cat# A5060) (Rabbit) from Sigma-Aldrich, 1:2,000 for anti-GFP antibody (Cat# sc-9996) (Rabbit) from Santa Cruz, 1:1,000 for anti-mCherry antibody (Cat# PA5-34974) (Rabbit) from Thermo Fisher Scientific, 1:1,000 for anti-HA antibody (Cat# 2367S) (Mouse) from CST and Normal Rabbit IgG (sc-2027) from Santa Cruz, 1:1,000 for anti-dPIP4K (generated for the laboratory by NeoBiolab).

#### *Immunoprecipitation*
About 2 million S2R+ cells were transfected for 48 h and lysates were prepared using 200 $\mu$l of same lysis buffer used for the preparation of protein samples for Western blotting. After lysis for 15 min at 4°C, the samples were spun down at 13,000$g$ for 10 min to remove cellular debris. 5% of the supernatant obtained was kept aside for input control; to the rest of the sample, lysis buffer was added to make up the volume to 1 ml. The volumes were split in two halves—one for IgG control and the other for immunoprecipitation. About 1.6 $\mu$g equivalent of antibody/normal rabbit IgG was used for overnight incubation at 4°C with continuous rotation. mCherry-tagged *Drosophila* Fab1 complexes with anti-mCherry antibody were precipitated by ~60 $\mu$l slurry of washed and blocked protein-G sepharose beads (according to the manufacturer's protocol, # GE17-0886-01; Sigma-Aldrich) for 2 h at 4°C. The beads were then washed with 0.1% TBST containing 0.1% 2-mercaptoethanol, 0.1mM EGTA two times and resuspended in 100 $\mu$l of the same buffer and stored at 4°C till the kinase assay was performed.

### Cell size measurements

Salivary glands were dissected from wandering third instar larvae and fixed in 4% paraformaldehyde for 20 min at 4°C. Post fixation, glands were washed thrice with 1X PBS and incubated in BODIPY FL C12-sphingomyelin (Cat# D7711) for 3 h at room temperature, after which, they were washed thrice in 1X PBS and stained with either DAPI (cat# D1306; Thermo Fisher Scientific) or TOTO3 (cat# T3604; Thermo Fisher Scientific) for 10 min at room temperature and washed

**The primers used were as follows.**

| Primer name | Sequence |
|---|---|
| Mtm Forward | 5'-TAGCCAGCAGTTCAACAACG-3' |
| Mtm Reverse | 5'-GTCTTGTGCTTGAGATCTTCGG-3' |
| CG3632 Forward | 5'-TGAAAAGGTTCTTTGGCCAGC-3' |
| CG3632 Reverse | 5'-CCATTGTGTCCGCTCTGTCT-3' |
| CG3530 Forward | 5'-TGGACACGTCGAGCTTCATC-3' |
| CG3530 Reverse | 5'-TCGGTAGTAGGGGTTCAGCA-3' |
| RP49 Forward | 5'-CGGATCGATATGCTAAGCTGT-3' |
| RP49 Reverse | 5'-GCGCTTGTTCGATCCGTA-3' |
| PI3K59F Forward | 5'-ACCTATTTGCTGGGTGTGGG-3' |
| PI3K59F Reverse | 5'-CCTTGCTCAGCTTCATTGGC-3' |
| PI3K68D Forward | 5'-CGAGGACTACTCCCGTGTGA-3' |
| PI3K68D Reverse | 5'-GTTGCTGCATCTCCGCTGTA-3' |
| UVRAG Forward | 5'-GCAAAGACATAAGGATGTTTTCG-3' |
| UVRAG Reverse | 5'-AATGTGAGGGGAGACAGAGG-3' |

with 1X PBS again. About 2–3 glands per slide were then mounted in 70% glycerol and imaged. Imaging was done on Olympus FV1000 or FV3000 Confocal microscope using a 10X objective. The images were then stitched into a 3D projection using an ImageJ plugin. These reconstituted 3D z stacks were then analyzed for nuclei numbers (for cell number) and volume of the whole gland using Volocity Software (version 5.5.1, Perkin Elmer Inc.). The average cell size was calculated as the ratio of the average volume of the gland to the number of nuclei.

## Cathepsin staining and analysis

Late feeding or early wandering third instar larvae were dissected in ice-cold Schneider's incomplete medium for the salivary glands and fixed in 4% PFA for 20 min at room temperature. Samples were rinsed twice in PBS and permeabilized for 15 min in PBTX-DOC (PBS with 0.1% Triton X-100 and 0.05% sodium deoxycholate). Blocking was done in 3% goat serum in PBTX-DOC for 3 h at room temperature and samples were then incubated overnight at 4°C with 1:200 anti-Cathepsin L/MEP antibody (ab58991; Abcam) in 1% goat serum in PBTX-DOC. The next day, after 3 × 30-min washes in PBTX-DOC, samples were then incubated with secondary antibody—goat anti-rabbit Alexa 488 (1:500) (A11034; Life Technologies) in 1% goat serum in PBTX-DOC for 4 h at room temperature. Finally, after 3 × 15 min washes in PBTX-DOC and staining for DAPI (1:1,000 for 10 min at room temperature), samples were mounted in 90% glycerol and imaged. Imaging was done on an Olympus FV3000 Confocal microscope. The acquired 3D images were stitched to give one 2D image using Zproject in ImageJ. The 2D images were then analysed for the size of punctae (analyzed using 3D object counter plugin in ImageJ). The size of lysosomes coming from one cell was summed and averaged to the number of lysosomes in that cell to give one value, that is, the average size of lysosomes in that cell. These 2D images were also analysed for the number of punctae using the 3D object counter plugin in ImageJ. The number of punctae were normalised to the area of the cell to give one value, that is, number of lysosomes per unit cell area.

## Atg8a punctae measurements

Around 40 first instar larvae were picked and incubated per vial to control for crowding. Salivary glands were dissected from wandering third instar larvae and fixed in 2.5% paraformaldehyde for 20 min at room temperature. Post fixation, glands were washed twice with 1X PBS. Glands were mounted in 70% glycerol and imaged on the same day. Imaging was done on an Olympus FV3000 Confocal microscope using a 60X objective. The 3D images were stitched to give one 2D image using Zproject in ImageJ. These 2D images were then analysed for the number of punctae (analysed using 3D object counter plugin in ImageJ). The number of punctae was normalised to the area of the cell and plotted for respective genotypes.

## 2XFYVE and LTS punctae measurements

Salivary glands from the corresponding genotypes were dissected, fixed and imaged as described for the Atg8a punctae measurements. The 3D images were stitched to give one 2D image using

Zproject in ImageJ. The 2D images were then analysed for the number of punctae (analysed using 3D object counter plugin in ImageJ). The number of punctae were normalised to the area of the cell and plotted for the respective genotypes.

## Lipid standards

diC16-PI3P – Echelon P-3016; diC16-PI4P – Echelon P-4016; Avanti 850172 | rac-16:0 PI(3,5)P$_2$-d5 (Custom synthesised); 17: 0 20: 4 PI3P - Avanti LM-1900; 17: 0 20: 4; PI(4,5)P$_2$ - Avanti LM-1904, 17:0 14:1 PI – Avanti LM-1504.

## Radioactivity-based PI3P mass assay

diC16-PI3P (Echelon) was mixed with 20 $\mu$M phosphatidylserine (PS) (#P5660; Sigma-Aldrich) and dried in a centrifugal vacuum concentrator. For biological samples, PS was added to the organic phase obtained at the end of the neomycin chromatography before drying. To the dried lipid extracts, 50 $\mu$l 10 mM Tris–HCl pH 7.4 and 50 $\mu$l diethyl ether were added and the mixture was sonicated for 2 min in a bath sonicator to form lipid micelles. The tubes were centrifuged at 1000$g$ to obtain a diethyl ether phase and vacuum centrifuged for 2 min to evaporate out the diethyl ether. At this time, the reaction was incubated on ice for about 10 min and 2X kinase assay buffer (100 mM Tris pH 7.4, 20 mM MgCl$_2$, 140 mM KCl, and 2 mM EGTA) and 10 $\mu$l immunoprecipitated dFab1 bead slurry was added. To this reaction, 10 $\mu$Ci [$\gamma$-$^{32}$P] ATP was added and incubated at 30°C for 16 h. Post 16 h, the lipids were extracted from the reaction as described earlier in a radioactive PI5P mass assay protocol (Jones et al, 2013).

## Thin-layer chromatography

The extracted lipids were resuspended in chloroform and resolved by TLC (preactivated by heating at 90°C for 1 h) with a running solvent (45:35:8:2 chloroform: methanol: water: 25% ammonia). Plates were air-dried and imaged on a Typhoon Variable Mode Imager (Amersham Biosciences).

## In vitro dFab1 immunoprecipitate-based lipid 5-kinase assays

600 pmol of either 17:0 | 14:1 PI (Avanti # LM 1504) or 17:0 | 20:4 PI3P (# LM 1900; Avanti) were mixed with 20 $\mu$l of 0.5 (M) of PS (P5660; Sigma-Aldrich) and dried in a centrifugal vacuum concentrator. To this, 50 $\mu$l 10 mM Tris–HCl pH 7.4 was added and the mixture was sonicated for 3 min in a bath sonicator to form lipid micelles. At this time, the reaction was incubated on ice for ~ 10 min and 2X kinase assay buffer (100 mM Tris pH 7.4, 20 mM MgCl2, 140 mM KCl, and 2 mM EGTA and equal volumes of immunoprecipitated dFab1 was added. For LC-MS/MS-based experiments the kinase assay buffer contained 80 $\mu$M cold ATP [10519979001; Roche]). The rest of the procedure was followed as mentioned in the following section.

## In vitro larval lysate-based lipid 3-phosphatase assays

The assay conditions have been adopted from a previous study (Schaletzky et al, 2003). The phosphatase assay comprises three parts—(i) preparation of lysate: third instar wandering larvae were

collected in groups of 5 or S2R+ cells treated with dsRNA were lysed in phosphatase lysis buffer containing 20 mM Tris–HCl (pH 7.4), 150 mM NaCl, 1% Triton X-100 (vol/vol) and protease inhibitor cocktail (Roche), by incubating the resuspended mixture in ice for 15–20 min. The larval carcasses were removed by a brief spin for 5 min at 1,000$g$ speed. Total protein was estimated by Bradford's reagent and desired amount of lysate was used for the subsequent assay. (ii) Lipid phosphatase assay: 600 pmol of either 17:0 20:4 PI3P or d5-PI(3,5)P$_2$ lipid was mixed and dried with 20 $\mu$l of 0.5 M bovine brain-derived PS (#P5660; Sigma-Aldrich) followed by bath sonication of the mixture in presence of 50 $\mu$l of 10 mM Tris–HCl (pH 7.4) for 3 min at maximum amplitude. To this, 50 $\mu$l of 2× phosphatase assay buffer (Schaletzky et al, 2003) and 10 $\mu$g total protein equivalents of cell-free lysate was added and the reaction was incubated for 15 min at 37°C. The reaction was quenched with 125 $\mu$l of 2.4 (N) HCl followed by lipid extraction described earlier (Jones et al, 2013). Samples for the PI3P assay were processed according to section (iii). For the PI(3,5)P2 phosphatase assay, the dried lipids from this step were resuspended in 20 $\mu$l of 0.5 M PS and dried. To this, 50 $\mu$l of 10 mM Tris–HCl (pH 7.4) was added and bath sonicated for 3 min similar to the first step of the assay. At this step, 50 $\mu$l of 2× kinase buffer containing 80 $\mu$M O$^{18}$ ATP (OLM-7858-PK; Cambridge Isotope laboratories, Inc.) and 1 $\mu$g of bacterial-purified human PIP4K$\alpha$-GST were added and the reaction was incubated at 30°C for 1 h. This was followed by lipid extraction as described in the previous step. (iii) Derivatization of lipids and LC-MS/MS: the organic phases were collected from the last step and dried and 50 $\mu$l of 2M TMS-diazomethane (#AC385330050; Acros) was added to each tube and vortexed gently for 10 min at room temperature. The reaction was neutralized using 10 $\mu$l of glacial acetic acid. The samples were dried in vacuo and 200 $\mu$l of methanol was used to reconstitute the sample to make it ready for injection for LC-MS/MS analysis.

## Lipid isolation for PI5P and PI3P measurements

Lipids from larvae were isolated from 3 or 5 third instar wandering larvae for PI5P or PI3P measurements, respectively. Total lipids were isolated and neomycin chromatography (for PI5P measurements only) was performed as described earlier (Ghosh et al, 2019). Briefly, five wandering third instar larvae were washed, dried on a tissue paper, and transferred to 0.5 ml tubes (KT03961-1-203.05; Precellys Bertin corp.) containing 200 $\mu$l phosphoinositide elution buffer (chloroform/methanol/2.4 M hydrochloric acid in a ratio of 250/500/200 [vol/vol/vol]). A Bertin homogenizer instrument, Precellys24 (P000669-PR240-A), was used at 8,000 rcf for four cycles with 30 s rest time on ice. The homogenate was transferred to 750 $\mu$l of phosphoinositide elution buffer in a 1.5-ml Eppendorf and sonicated for 2 min. We added either 10 or 35 ng of 17:0 20:4 PI(4,5)P$_2$ as internal standard (in methanol) for LC-MS/MS experiments. Furthermore, 250 $\mu$l chloroform and 250 $\mu$l MS-grade water were added and vortexed for 2 min. The contents were then centrifuged for 5 min at 1,000$g$ to obtain clean phase separation. The lower organic phase was washed with equal volume of lower phase wash solution (methanol/1 M hydrochloric acid/chloroform in a ratio of 235/245/15 [vol/vol/vol]) and vortexed and phase separated. The organic phase thus obtained was dried in vacuum and stored at -20°C and

processed further within 24 h. For the PI5P mass assay, phosphoinositides were enriched using neomycin chromatography. Neomycin beads were purchased from Echelon Biosciences (cat. no. P-B999). Briefly, batch purification was performed with 1–2 mg bead equivalent in slurry form on a Rotospin instrument (Tarsons) using buffers as described in Jones et al [2013].

## LC-MS/MS for in vitro assays and PI5P measurements

The instrument operation was followed similar to the description in our previous methods work on PI5P quantification (Ghosh et al, 2019). For in vivo lipid measurements, the samples were washed with post-derivatisation wash step before injecting in mass spec. Samples were run on a hybrid triple quadrupole mass spectrometer (Sciex 6500 Q-Trap or Sciex 5500 Q-Trap) connected to a Waters Acquity UPLC I class system. Separation was performed on a ACQUITY UPLC Protein BEH C4, 300 Å, 1.7 $\mu$m, 1 mm X 100 mm column (Product #186005590) using a 45–100% acetonitrile in water (with 0.1% formic acid) gradient over 10 min. MS/MS and LC conditions used were as described earlier (Ghosh et al, 2019).

## Larval PI3P measurements

We adopted a previously used method of deacylation of total lipids followed by detection by LC-MS/MS using ion-paring-based separation chemistry followed by detection using mass spec (Kiefer et al, 2010; Jeschke et al, 2015). Using our conditions, we could not reproducibly separate the deacylated PI5P isomeric peak from biological samples. But we could always separate deacylated PI3P from PI4P in these biological samples (Fig 3B). Synthetic standards were used to determine the retention times ($R_t$) of the individual peaks. Fig S6A shows synthetic GroPI3P at $R_t$ = 6.13 min and GroPI4P at $R_t$ = 6.95 min. The $R_t$ of GroPI3P and GroPI4P was shifted in case of biological samples and to confirm the peaks were representative of the corresponding analytes, we spiked synthetic GroPI3P into the biological sample of Fig S6B. As expected, we observed a spike in the first peak, albeit at $R_t$ = 7.65 min, without a significant change in the second peak at $R_t$ = 8.70 min, thus confirming that the first peak was indeed corresponding to PI3P. Furthermore, we also verified that GroPI3P can be linearly detected at a range of 30–3,000 pg on column and GroPI4P can be linearly detected at a range of 30–4,000 pg on column (Fig S6C and D). We determined that the limit of detection was 20 pg on column for GroPI3P and GroPI4P as concluded from signal to noise (S/N) being 30 and 24, respectively.

The following are the steps by which PI3P measurements were performed: (i) larval lipid extraction: as mentioned in previous section; (ii) lipid deacylation: dried lipid extracts were incubated with 1 ml of 25% methylamine solution in water/methanol/n-butanol (43:46:11) at 60°C for 1 h followed by drying this extract in vacuo (~3–4 h); (iii) fatty acid wash: next, the lipids were reconstituted in 40–50 $\mu$l MS-grade water, and to this, an equal volume fatty acid extraction reagent (1-butanol/petroleum ether [40–60°C boiling]/ethyl formate in a ratio of 20/4/1 [vol/vol/vol]) was added and vortexed for 2 min. Following this, the tubes were centrifuged for 5 min at 1,000$g$ to obtain phase separation. The upper organic phase was discarded, and the lower aqueous phase was processed for LC-MS/MS analysis.

**Multiple reaction monitoring (MRM) values for commercial lipids used.**

| Lipids | Parent ion | Daughter ion |
|---|---|---|
| d5-diC16-PI5P | 938.5 | 556.5 |
| d5-diC16-PI(3,5)$P_2$ | 1046.5 | 556.5 |
| d5-diC16-$^{18}$O-$PIP_2$ | 1052.5 | 556.5 |
| 17:0| 20:4 PI3P | 995.5 | 613.5 |
| 17:0| 20:4 $PIP_2$ | 1103.5 | 613.5 |
| 17:0|14:1 PI | 809.4 | 535.4 |
| 17:0|14:1 PIP | 917.4 | 535.4 |

### LC-MS/MS for deacylated PI3P measurements

Deacylated PIPs (GroPI3P and GroPI4P) were run on a hybrid triple quadrupole mass spectrometer (Sciex 6500 Q-Trap) connected to a Waters Acquity UPLC I class system. Separation was performed on a Phenomenex Synergi 2.5 $\mu$m Fusion-RP 100 Å, LC Column 100 × 2 mm, (Product # 00D-4423-B0) maintained at 32°C during the run. Mobile phase A consisted of 4 mM DMHA and 5 mM acetic acid in water, and mobile phase B consisted of 4 mM DMHA and 5 mM acetic acid in 100% methanol. Flow rate was 0.2 ml/min.

The process of linear gradient elution was conducted as follows: 0–2 min (methanol, 3%), 2–5 min (methanol, 7%), 5–8 min (methanol, 12%), and 8–9 min (methanol, 100%). For the next 4 min, solvent B was maintained at 100%. Then, equilibration was performed between 12.2 and 20.0 min using 3% methanol. The injection volume and running time of each sample was 3.0 $\mu$l and 20.0 min, respectively.

Mass spectrometry data were acquired in MRM mode in negative polarity. Quantification of PIPs was achieved with the MRM pair (Q1/Q3) m/z 413→259. Electrospray (ESI) voltage was at − 4,200 V and TEM (Source Temperature) as 350°C, DP (Declustering Potential) at −55, EP (Entrance Potential) at −10, CE (Collision Energy) at −31, CXP (Collision cell Exit Potential) at −12. Dwell time of 100 milliseconds was used for experiments with CAD value of −3, GS1 and GS2 at 25, CUR (Curtain gas) at 40. Both Q1 and Q3 masses were scanned at unit mass resolution.

### Total organic phosphate measurement

500 $\mu$l flow-through obtained from the phosphoinositide-binding step of neomycin chromatography was used for the assay for measurements of PI5P. For PI3P measurements, 50 $\mu$l was obtained from the last step of lipid extraction and stored separately in phosphate-free glass tubes till the assay was performed. The sample was heated till drying in a dry heat bath at 90°C in phosphate-free glass tubes (Cat# 14-962-26F). The rest of the process was followed as described previously (Jones et al, 2013).

### Software and data analysis

Image analysis was performed by Fiji software (Open source). Mass spec data were acquired on Analyst 1.6.2 software followed by data processing and visualisation using MultiQuant 3.0.1 software and PeakView Version 2.0., respectively. Chemical structures were drawn with ChemDraw Version 16.0.1.4. Illustrations were created with BioRender.com. All datasets were statistically analysed using MS Excel (Office 2016) and GraphPad Prism 9.

# Supplementary Information

# Acknowledgements

This work was supported by the Department of Atomic Energy, Government of India, under Project Identification No. RTI 4006 and a Wellcome-DBT India Alliance Senior Fellowship to P Raghu (IA/S/14/2/501540). We thank the NCBS Imaging and Mass Spectrometry Facility and the *Drosophila* facility for support. We thank Rajan Thakur and members of the Padinjat laboratory for help and advice during this study.

## Author Contributions

A Ghosh: conceptualization, resources, formal analysis, validation, investigation, visualization, methodology, and writing—original draft, review, and editing.
A Venugopal: data curation, formal analysis, validation, investigation, visualization, methodology, and writing—original draft, review, and editing.
D Shinde: methodology.
S Sharma: investigation and methodology.
M Krishnan: investigation and methodology.
S Mathre: investigation and methodology.
H Krishnan: investigation and methodology.
S Saha: investigation and methodology.
P Raghu: conceptualization, supervision, project administration, and writing—original draft, review, and editing.

## Conflict of Interest Statement

The authors declare that they have no conflict of interest.

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
