## [Reviewer comments · Life Science Alliance]

Life Science Alliance

PI3P dependent regulation of cell size and autophagy by phosphatidylinositol 5-phosphate 4-kinase

Avishek Ghosh, Aishwarya Venugopal, Dhananjay Shinde, Sanjeev Sharma, Meera Krishnan, Swarna Mathre, Harini Krishnan, Padinjat Raghu, and Sankhanil Saha

DOI: <https://doi.org/10.26508/lsa.202301920>

Corresponding author(s): *Padinjat Raghu, National Centre for Biological Sciences*

Review Timeline:

Submission Date:	2023-01-12
Editorial Decision:	2023-01-12
Revision Received:	2023-05-22
Editorial Decision:	2023-05-26
Revision Received:	2023-06-02
Accepted:	2023-06-02

Scientific Editor: *Eric Sawey, PhD*

Transaction Report:

Please note that the manuscript was reviewed at Review Commons and these reports were taken into account in the decision-making process at *Life Science Alliance*.

Review
COMMONS

January 12, 2023

Re: Life Science Alliance manuscript #LSA-2023-01920

Prof. Raghu Padinjat
National Centre for Biological Sciences
Cellular Organization and Signalling
TIFR GKVK Campus
Bangalore, Karnataka 560065
India

Dear Dr. Padinjat,

Thank you for submitting your manuscript entitled "PI3P dependent regulation of cell size and autophagy by phosphatidylinositol 5-phosphate 4-kinase" to Life Science Alliance. We invite you to re-submit the manuscript, revised according to your Revision Plan.

Thank you for this interesting contribution to Life Science Alliance. We are looking forward to receiving your revised manuscript.

Sincerely,

B. MANUSCRIPT ORGANIZATION AND FORMATTING:

Manuscript number: RC- 2022-01627

Corresponding author(s): Padinjat, Raghu

[The “revision plan” should delineate the revisions that authors intend to carry out in response to the points raised by the referees. It also provides the authors with the opportunity to explain their view of the paper and of the referee reports.]

The document is important for the editors of affiliate journals when they make a first decision on the transferred manuscript. It will also be useful to readers of the reprint and help them to obtain a balanced view of the paper.

*If you wish to submit a full revision, please use our "Full Revision" template. **It is important to use the appropriate template to clearly inform the editors of your intentions.**]*

1. General Statements [optional]

PIP4K are a relatively new class of phosphoinositide kinases that are specific to metazoan genomes. Many genetic studies in model organisms have underscored the importance of these enzymes in key physiological process. However, there is a lack of understanding of the biochemical mechanism by which PIP4K enzymes regulate physiological processes.

Previous studies have focused on the likely importance of PIP4K in converting PI5P into PI(4,5)P₂. In this study, we discover a potential new mechanism by which PIP4K could regulate one physiological process in *Drosophila*, i.e., the regulation of cell size.

We thank all reviewers for their detailed and constructive reviews. The questions raised by the reviewers are amongst the most challenging in the field. However, having considered the reviewer comments, we propose revisions to text and additional experimental data that should address most of the points raised by reviewer comments. With these revisions, we believe our manuscript will be an important new advance to the field of PIP4K but also the areas of growth control, autophagy, and endo-lysosomal homeostasis.

Description of the planned revisions

Based on the comments of all three reviewers, there are two key issues that are highlighted in respect of the scientific message of this manuscript.

- (1) A need to provide more evidence on the model that PIP4K may be modulating PI3P levels in the context of the autophagosome pathway.**

(2) Is the dPIP4K enzyme regulating PI3P levels directly (i.e. by mediating phosphorylation of PI3P to generate PI(3,4)P₂) OR is the PI3P levels being controlled indirectly by regulating other lipid kinases that may phosphorylate or dephosphorylate PI3P.

Revisions including new experiments are listed below. In addition to the revisions related to points 1 and 2 above we have also listed other miscellaneous revisions not related to 1 and 2 as a separate section below.

(1) A need to provide more evidence that PIP4K may be modulating PI3P levels in the context of the autophagosome pathway.

Figure R0: (A) (i) Representative confocal z-projections depicting lysosomes using LTS::mCherry in the salivary glands from the genotypes a. *AB1>LTS::mCherry* and b. *AB1>LTS::mCherry; Fab1 RNAi*. Scale bar indicated at 20 μ m. (ii) Graph representing measurement of LTS::mCherry punctae numbers per unit cell area of the salivary glands from wandering third instar larvae of *AB1>LTS::mCherry* (N=3, n=15) and *AB1>LTS::mCherry; Fab1i* (N=3, n=15). Student's unpaired t-test with Welch correction showed p value < 0.0001. (iii) Graph representing measurement of average size of LTS::mCherry punctae per unit cell of the salivary glands from wandering third instar larvae of *AB1>LTS::mCherry* (N=3, n=15) and *AB1>LTS::mCherry; Fab1i* (N=3, n=15). Student's unpaired t-test with Welch correction showed p value = 0.0104. (B) (i) Representative confocal z-projections depicting lysosomes using LTS::mCherry in the salivary glands from the genotypes a. *AB1>LTS::mCherry* and b. *AB1>LTS::mCherry; dPIP4K²⁹*. Scale bar indicated at 20 μ m (ii) Graph representing measurement of LTS::mCherry punctae numbers per unit cell area

of the salivary glands from wandering third instar larvae of AB1>LTS::mCherry (N=5,n=25) and AB1>LTS::mCherry; *dPIP4K²⁹* (N=2,n=25). Student's unpaired t-test with Welch correction showed p value = 0.2591. (iii) Graph representing measurement of average size of LTS::mCherry punctae per unit cell of the salivary glands from wandering third instar larvae of AB1>LTS::mCherry (N=5,n=25) and AB1>LTS::mCherry; *dPIP4K²⁹*(N=5,n=25). Student's unpaired t-test with Welch correction showed p value = 0.0049. (C) (i) Representative confocal z-projections depicting lysosomes using Cathepsin L staining in the salivary glands from the genotypes a. *w¹¹¹⁸* (Control) and b. *dPIP4K²⁹* (mutant). Scale bar indicated at 20 μm (ii) Graph representing measurement of Cathepsin L punctae numbers per unit cell area of the salivary glands from wandering third instar larvae of *w¹¹¹⁸* (Control) and *dPIP4K²⁹* (mutant). Student's unpaired t-test with Welch correction showed p value = 0.2900. (iii) Graph representing measurement of average size of Cathepsin L punctae per unit cell of the salivary glands from wandering third instar larvae of *w¹¹¹⁸* (Control) and *dPIP4K²⁹* (mutant). Student's unpaired t-test with Welch correction showed p value <0.0001.

To determine any changes in the autophagosome, lysosome system in relation to autophagy, we generated transgenic flies expressing the probe LTS::mcherry where mcherry is fused in frame with an N-terminal lysosomal targeting sequence(Figure R0 A); this probe is targeted to lysosomes and can be used for visualising this organelle in cells. To test this probe we expressed it in salivary glands where the enzyme Fab1, a 5-kinase that regulates late endosome-lysosome/lysosome-lysosome/autophagosome-lysosome fusion had been depleted using RNAi. In these cells, depletion of Fab1 results in a reduction in the number of lysosomes and an increase in the average size of lysosomes [Figure R0 A(ii) and A(iii)]. as previously reported using other lysosomal probes (Rusten et al., 2006) Using this probe, we found no reduction in the number of lysosomes in *dPIP4K²⁹* (Figure R0 B(ii)). Independently, we also tested the status of active lysosomes using an antibody to cathepsin L, a lysosomal endopeptidase. Here too, we found no difference in the number of Cathepsin L positive punctate in *dPIP4K²⁹* [Figure R0 C(ii)]. Together these findings argue that while there is enhanced autophagy in *dPIP4K²⁹* (as assessed by Atg8a (Figure 6Ai-ii)), there may also be defects in the later processing of Atg8a autophagosomes as they reach the lysosomal compartment that needs further evaluation.

- Autophagosome-Lysosome fusion defect

Atg8a fused to GFP and mCherry (so called Traffic light construct) has been used in mammalian cells to detect a defect in autophagosome/lysosome fusion. In this assay, yellow punctae corresponds to the autophagosomes while red punctae corresponds to the autolysosomes. This approach has also been reported in the *Drosophila* system and we have already attempted to carry out this assay. However, we faced technical difficulties using the traffic light construct in *Drosophila* salivary gland cells. Much to our surprise, we could not see red punctae at all in salivary glands; the reason for this is unclear but it precludes us from performing this specific experiment.

As an alternative, we performed an *ex vivo* Bafilomycin A1 (BafA1) treatment assay on freshly dissected salivary glands. By comparing BafA1 sensitivity in wild type versus *dPIP4K²⁹*, we tested whether autophagosome flux is altered. To perform this experiment, we dissected salivary glands from wild type controls (*w¹¹¹⁸*) and *dPIP4K²⁹* glands in Schneider's incomplete medium with with human insulin (10 ug/ml) and treated with or without 500 nM Bafilomycin A1(BafA1) in DMSO for 3 hours. Post treatment, the glands were fixed and stained with Atg8a antibody (gift from Rachel Kraut). Firstly, we observed an increase in Atg8a structures in control cells with BafA1 treatment as compared to DMSO treatment (Fig R1A; quantified in Fig R1B) confirming that the BafA1 treatment paradigm worked under our conditions.

However, upon observing *dPIP4K²⁹* cells, we found a diffused staining pattern in cells treated with BafA1 as compared to cells treated with DMSO. This staining was refractory to the 3D object counter program that we have used to quantify other endomembrane structures in this study. The diffused staining could arise from either an artefactual non-specific signal or as a function of relatively small but higher number of newly formed autophagosomes which do not have detectable antigenicity for Atg8a above the background level for the 3 hours BafA1 treatment. Therefore, at present, we cannot make a definitive conclusion on whether dPIP4K affects autophagosome-lysosome fusion in salivary glands. In future, we would like to optimize this experiment further specifically with further validation of the Atg8a antibody.

Taken together these lines of evidence suggest that in these cells, the loss of dPIP4K (i) a reduction in the number of lysosome structures as reported by LTS::mcherry punctae (ii) the number of active lysosome structures as reported by cathepsin-L punctae is not reduced between control and *dPIP4K²⁹*. There seems to be differences in the morphology of lysosome related structures in *dPIP4K²⁹* which are not fully understood and will need to be studied going forwards.

[Figure removed by editorial staff per authors' request]

- Does PIP4K localize to the autophagosome compartment ?
This is a pertinent question. However, it has not been possible to assay for the localization of endogenous PIP4K in *Drosophila* (or any model system so far) due to lack of antibodies that are able to detect the endogenous protein in immunofluorescence studies. Therefore, we attempted colocalization studies with dPIP4K::eGFP and Atg8a::mCherry (Figure R2); we could not detect co-localization of these two proteins in salivary gland cells. It is possible that the localization of dPIP4K to the autophagosomal compartment is transient and not captured readily under our experimental conditions.

Figure R2: Representative confocal z-projections from 3rd instar *Drosophila* salivary glands showing the co-localization of (a) dPIP4K::eGFP , (b) Atg8a::mCherry and (c) Merged image of the two. Scale bar indicated at 20 μ m

(2) Is the dPIP4K enzyme regulating PI3P levels directly (i.e by mediating phosphorylation of PI3P to generate PI(3,4)P₂ OR is the PI3P levels being controlled indirectly by regulating other lipid kinases that may phosphorylate or dephosphorylate PI3P.

We have previously reported that purified dPIP4K has *in vitro* activity on PI3P to produce PI(3,4)P₂ (Gupta et al., 2013). We and others have also reported that the human PIP4K α also shows activity on PI3P to produce PI(3,4)P₂ *in vitro* (Zhang et al. 1997; Gupta et al. 2013). Therefore, the ability of PIP4K to phosphorylate PI3P *in vitro* has been previously demonstrated by more than one group. The question is whether this *in vitro* activity on PI3P is relevant *in vivo*. In this paper, we report that PI3P levels are elevated in dPIP4K mutants, and that this elevation can be reverted by a wild type dPIP4K transgene but not a kinase dead one. This shows that the kinase activity of PIP4K is required to regulate PI3P levels *in vivo*.

“If the authors can show the activity in flies is real by measuring the product PI34P2 this would be compelling evidence.”

An independent way of testing whether dPIP4K can phosphorylate PI3P *in vivo* is to measure the levels of the product, namely PI(3,4)P₂ that would be generated by this reaction. Therefore, we agree that a direct measurement of PI(3,4)P₂ levels from larval tissues of *dPIP4K²⁹* would be valuable in the current study. However, without radiolabelling, deacylation and ion exchange chromatography, PI(3,4)P₂ measurements are difficult to perform, primarily due to the very low levels of this lipid. Therefore, it is a significant challenge to do this, biochemically in flies compared to mammalian cell lines.

Previous studies have reported the use of a genetically encoded protein sensor called TAPP1 protein fused to a fluorophore to detect PI(3,4)P₂. Recently Goulden et.al (Goulden et al., 2019), came up with a modified version of this sensor, wherein the c terminal PH domain of the TAPP1 protein is repeated thrice in tandem to give cPHx3. cPHx3 fused to a fluorophore is a high avidity biosensor to detect changes in PI(3,4)P₂. This probe has not been used in the *Drosophila* system previously. Hence, we generated transgenic flies expressing

cPHx3::eGFP [Figure R3 A(i)]. In wild type cells most of this PI(3,4)P₂ probe is distributed at the plasma membrane [Figure R3A(ii)] with very few punctae in cells, indicating an extremely low level of endomembranous pool of PI(3,4)P₂ in these cells under resting conditions. To address this concern, we elevated PI3P levels using *mtm* RNAi. If PI3P is a relevant substrate of PIP4K then elevation of PI3P levels could result in an increase of PI(3,4)P₂ levels via the activity of PIP4K. We tested this hypothesis by measuring PI(3,4)P₂ levels using cPHx3::GFP and found several PI(3,4)P₂ punctae distributed throughout the cell body [Figure R3B(i)] quantified in Figure R3B(ii). This system can be used in the future to address in detail the impact of manipulating dPIP4K in generating PI(3,4)P₂ from PI3P.

[Figure removed by editorial staff per authors' request]

- **Figure 4D: Does the A381E mutant of PIP4K affect PI3P levels in cells as it cannot reverse the cell size phenotype in Figure S1B?**

This is an interesting question raised by Reviewer #2. The hPIP4KB2 [A381E] has been reported to be a PI4P metabolizing enzyme due to a mutation in the C-terminal activation loop of PIP4K) (Kunz et al., 2002).

However, that study did not test its ability to phosphorylate PI3P. Hence, further characterization is required to understand if hPIP4KB2 [A381E] is active on PI3P.

Since measurement of PI3P levels from larval lipid extracts is non-trivial, we determined the *in vitro* activity of hPIP4KB2 and hPIP4KB2 [A381E] on PI3P as compared to PI5P. However, we could not observe a significant *in vitro* activity from our cell free lysate experiments for even the wild type hPIP4KB2. Therefore, to check if A381E of hPIP4K confers any activity to PI3P, we used hPIP4KA2 and its corresponding switch mutant, hPIP4KA2 [A371E] for a proof of concept experiment. We observe significantly high activity of cell lysates transfected with hPIP4KA2 on PI5P and a complete abolishment of activity on PI5P for hPIP4KA2 [A371E] as compared to untransfected control (UTC) lysates (Figure R4A(ii)). In contrast, neither the hPIP4KA2 nor hPIP4KA2 [A371E] showed significantly different activity on PI3P as compared to UTC lysates (Figure R4A(ii)). Taken together, we conclude that the hPIP4KB2 [A381E] which is in concept, a very similar switch mutant to hPIP4KA2 [A371E], will not have a regulatory effect on PI3P levels *in vivo*.

Figure R4: (A) (i) Immunoblot from *Drosophila* S2R+ cells showing the expression of hPIP4K2A::eGFP and hPIP4K2A A371E::eGFP. Actin was used as the loading control. (ii) *In vitro* kinase assay on synthetic PI5P and PI3P. Graph representing the normalised response ratio of “PI3P 4-kinase activity on PI3P” to “PI5P 4-kinase activity on PI5P” upon enzymatic activity of S2R+ cell free lysates expressing either no transgene (UTC), hPIP4K2A (2A) or hPIP4K2A[A371E]. Response ratio of “PI3P 4-kinase activity on PI3P” is obtained from area under the curve (AUC) of 17:0 20:4 PIP₂ (Product)/ 17:0 20:4 PI3P (Substrate), Response ratio of “PI5P 4-kinase activity on PI5P” is obtained from area under the curve (AUC) of 17:0 20:4 PIP₂ (Product)/ 17:0 20:4 PI5P (Substrate) and is represented as mean \pm S.E.M. Number of lysates = 2.

Figure 4G: The conclusion on line 255 that all phosphatase transcripts are unchanged in this figure when two of them appear to have significant reduction appears inaccurate. In addition, changes of transcript levels of these enzymes may not necessarily reflect their overall activity in cells. A localised reduction in MTM levels or activity may well play a role in dPIP4K29 cells even though an overall phosphatase activity is seen increased in the in vitro assay in Figure 4F. Similarly, not clear that the authors can completely rule out a potential activation of PIP3K59/vps34 and subsequent increase in PI3P levels in cells by simply looking at RNA levels. Is there a reason why the authors could not measure the enzyme levels in cells as mentioned in the text? VPS34 activity can be measured in mammalian systems. This is important as PI3PK59 KD does seem to reverse change in cell size (Figure 5A).

Response:

“The conclusion on line 255 that all phosphatase transcripts are unchanged in this figure when two of them appear to have significant reduction appears inaccurate. “

We thank Reviewer #2 for raising this point.

In fact the statement we have made in the manuscript is somewhat different. What we have stated is “In addition, we measured transcript levels of three putative 3-phosphatases – *Mtm*, *CG3632* and *CG3530* and found that the transcript levels of all the 3-phosphatases were unchanged in *dPIP4K²⁹* as compared to controls, **although there was an overall trend of decrease in all the genes** (Figure 4G).”

We have now included the actual p-values for the statistical test used to measure the significance of differences in the transcript levels for each phosphatase between wild type and dPIP4K mutants. As can be seen the difference in transcript levels for each phosphatase between wild type and mutants is modest and we leave it to the reader to infer if the difference is significant or not. “ In the light of the enhanced PI3P phosphatase activity we have observed in dPIP4K mutants (Figure 4E) it is not possible to make an unequivocal conclusion. This is stated in the text of the manuscript.

“A localised reduction in MTM levels or activity may well play a role in dPIP4K29 cells even though an overall phosphatase activity is seen increased in the in vitro assay in Figure 4F. “

This is a possibility but at the moment there is not a way to measure MTM activity with spatial resolution in cells and hence it is not possible to invoke this idea.

“Similarly, not clear that the authors can completely rule out a potential activation of PIP3K59/vps34 and subsequent increase in PI3P levels in cells by simply looking at RNA levels. Is there a reason why the authors could not measure the enzyme levels in cells as mentioned in the text? VPS34 activity can be measured in mammalian systems. This is important as PI3PK59 KD does seem to reverse change in cell size (Figure 5A).”

We agree with the reviewer on the general principle that mRNA levels need not reflect the impact on protein levels or its activity; this will also be true for the case of PI3K59F/Vps34. As a result, we have already spent considerable time and effort trying to set up Class III PI3K activity assays. We have tried a micelle-based PI preparation (similar to all the other in vitro kinase/phosphatase assays in our study) to standardize an assay for PI3K59F activity using total lysates from larvae but could not detect any significant 3-kinase activity in wild type controls. Further, we also tested lysates over-expressing PI3K59F^{3XHA} but again failed to detect significant activity. Since Class III PI3K is a multi subunit enzyme, this could be due to a lack of enrichment of the Vps34/Vps15 complex in the lysates, required for proper catalytic activity. Therefore, this assay could not be used to detect changes in PI-3-kinase activity in *dPIP4K²⁹* lysates. We may not be able to solve this issue easily, right now.

Another method to test the involvement of PI3K59/Vps34 is to target its adaptor proteins. Can the authors distinguish the endosomal and autophagosomal PIP3K59/vps34 complex and PI3P production by looking at drosophila homologues of Atg14 and UVRAG? The majority of PI3P in mammalian cells is found in the endosomal compartment rather than autophagosomal vesicles. If the authors predict that only autophagosomal PI3P levels are changed, then an overall change in enzymatic activity required for PI3P accumulation may not be easy to detect in total cell extracts.

Response – This is an excellent suggestion from the reviewer, and we have attempted it using RNAi depletion of UVRAG and Atg14L in dPIP4K mutant salivary glands. When UVRAG was depleted in dPIP4K mutant salivary glands (RNA knockdown of UVRAG was measured in whole larvae, Figure R5A), there was no effect on the cell size phenotype in either wild type or dPIP4K mutants (Figure R5B(i) and (ii)). However, the same UVRAG RNAi line resulted in a reduction in the endosomal pool of PI3P as measured by a 2XFYVE::GFP probe (Figure R6A(iii) and (iv)). These findings collectively indicate that the endosomal pool of PI3P regulated by the UVRAG/Vps34 complex is unlikely to regulate cell size in the salivary gland.

We also attempted to deplete Atg14L by RNAi. This did not reveal a reduction in cell size in wild type or dPIP4K mutants [Figure R5D(i) and (ii)]. However, Q RTPCR analysis revealed that the RNAi lines available to us gave only 45% (Atg14L) transcript depletion (Figure R5C). This may well be not enough to reduce the levels of Atg14L to the extent that it affects the function of this protein in cells.

Figure R5: (A) qPCR measurements for mRNA levels of *UVRAG* from either *da/+* (green) or *da> UVRAG RNAi* (magenta). Student's unpaired t-test showed p value = 0.0046. (n = 6, where each n is derived from 5 third instar wandering larvae) (B) (i) Graph representing average cell size measurement as mean \pm S.E.M. of salivary glands from wandering third instar larvae of *AB1/+* (n = 11) and *AB1> UVRAG RNAi* (n = 13). Student's unpaired t-test with Welch correction showed p value = 0.0882. (ii) Graph representing average cell size measurement as mean \pm S.E.M. of salivary glands from wandering third instar larvae of *AB1/+ ; dPIP4K²⁹* (n = 12) and *AB1> UVRAG RNAi ; dPIP4K²⁹* (n = 12). Student's unpaired t-test with Welch correction showed p value = 0.4306. (C) qPCR measurements for mRNA levels of *Atg14L* from either *da/+* (green) or *da> Atg14L RNAi* (magenta). Student's unpaired t-test showed p value < 0.0001. (n = 6, where each n is derived from 5 third instar wandering larvae) (D) (i) Graph representing average cell size measurement as mean \pm S.E.M. of salivary glands from wandering third instar larvae of *AB1/+* (n = 11) and *AB1> Atg14L RNAi* (n = 13). Student's unpaired t-test with Welch correction showed p value = 0.3851. (ii) Graph representing average cell size measurement as mean \pm S.E.M. of salivary glands from wandering third instar larvae of *AB1/+ ; dPIP4K²⁹* (n = 12) and *AB1> Atg14L RNAi ; dPIP4K²⁹* (n = 12). Student's unpaired t-test with Welch correction showed p value = 0.6543. Sample size is also represented by points on individual bars.

(3) Revisions related to other miscellaneous comments

It's not clear why there are no differences in PI3P/PIP₂ levels in Figure 4B, but this is overcome by normalizing to organic phosphate levels (4C)? Can differences in PI3P/PIP₂ levels be seen in Figure 4B without normalization if additional controls such as PI3K59F/VPS34 KD were used (as done in figure 5B)? A discussion of this could be useful.

Response: We thank Reviewer #2 for this comment. We acknowledge that the TLC image shown in Figure 4B shows almost similar PIP₂ spot intensities for various genotypes. When extracting lipids for this experiment from samples of individual genotypes, there is often differences in the recovery of total lipids from individual samples; this cannot be entirely avoided even by experienced workers. These differences in lipid extraction are handled by also measuring total organic phosphate from each extracted sample and then normalizing the levels of a desired lipid to the total amount of organic phosphate in that sample. Specifically, the organic phosphate levels of *dPIP4K²⁹* mutant larvae are lower than that of wild type controls. This is reflective of the previously reported smaller size of the larvae of *dPIP4K²⁹* mutant. In the quantification of PI3P levels shown in Figure 4A the intensity of the PIP₂ spot has been normalized to the total organic phosphate level in each sample.

As pointed out by Reviewer #2, we have discussed this in the text and will move the TLC figure to the supplementary data.

Figure 5C&D: how specific is the FYVE domain fused probes to endosomal PI3P? Such probes are used in mammalian cells to measure overall PI3P, whether endosomal or autophagosomal. In addition, such probes when expressed in live cells can alter PI3P generation. In line with this comment, FYVE-domain probes can be used to quantify PI3P levels in fixed cells, this method could be used to verify changes in PI3P levels seen in PIP4K mutant flies.

Response - This is a very interesting point. We do agree that 2XFYVE as a probe being specific to endosomal PI3P has not been tested in the salivary glands of *Drosophila*. We expressed Vps34, Atg1 and UVRAG RNAi using salivary gland specific Gal4 (AB1Gal4) and checked for the levels of 2XFYVE probe. Also, as a complementary experiment, we used these very same RNAi lines to check for the levels of Atg8a-mCherry in salivary glands.

We have experimentally tested the ability of the 2XFYVE probe to report endosomal and autophagosomal pools of PI3P in *Drosophila* salivary gland cells. Briefly, we depleted Vps34 and found that the total number of 2XFYVE punctae was reduced (Figure R6A(i) and (ii)). This was also the case when UVRAG, a component of the endosomal Vps34 complex II was depleted (Figure R6A(iii) and (iv)). However, when Atg1, an autophagy initiating kinase, was depleted we found no change in the number or intensity of 2XFYVE punctae (Figure R6A(v) and (vi)). Under these same conditions, depletion of Atg1 was able to reduce the number of Atg8a punctae in these cells (Figure R6B(v) and (vi)). However, Vps34 is also part of the Vps34 Complex I which is responsible for production of PI3P at the phagophore membrane. Therefore, we observed a significant reduction of Atg8a punctae upon knockdown of Vps34 in salivary glands (Figure R7B(i) and (ii)). In contrast, UVRAG RNAi did not cause a drastic decrease in Atg8a punctae under the same conditions (Figure B(iii) and (iv)). These findings strongly suggest that in salivary gland cells, the 2XFYVE probe

measurements in our assay is not able to report PI3P levels in autophagosomes. We are not aware of any published tool where a PI3P probe can be selectively targeted to measure autophagosomal PI3P at basal conditions.

Figure R6: (A) (i) Representative confocal z-projections depicting 2XFYVE punctae using 2XFYVE-mCherry in the salivary glands from the genotypes a. AB1>2XFYVE-mCherry, b. AB1>2XFYVE-mCherry; Vps34 RNAi. (ii) Graph representing measurement of 2XFYVE punctae numbers per unit cell area of the salivary glands from wandering third instar larvae of AB1>2XFYVE-mCherry (N=5, n =25), AB1>2XFYVE-mCherry ; Vps34 RNAi (N=5,n =25). Student's unpaired t-test with Welch correction showed p value<0.0001.(iii) Representative confocal z-projections depicting 2XFYVE punctae using 2XFYVE-mCherry in the salivary glands from the genotypes a. AB1>2XFYVE-mCherry, b. AB1>2XFYVE-mCherry; UVRAG RNAi. (iv) Graph representing measurement of 2XFYVE punctae numbers per unit cell area of the salivary glands from wandering third instar larvae of AB1>2XFYVE-mCherry (N=3, n =11), AB1>2XFYVE-mCherry ; UVRAG RNAi (N=3,n =11). Student's unpaired t-test with Welch correction showed p value<0.0001 (v) Representative confocal z-projections depicting 2XFYVE punctae using 2XFYVE-mCherry in the salivary glands from the genotypes a. AB1>2XFYVE-mCherry, b. AB1>2XFYVE-mCherry; Atg1 RNAi. (vi)

Graph representing measurement of 2XFYVE punctae numbers per unit cell area of the salivary glands from wandering third instar larvae of AB1>2XFYVE-mCherry (N=5, n = 20), AB1>2XFYVE-mCherry ; Atg1 RNAi (N=5,n =20). Student's unpaired t-test with Welch correction showed $p=0.0946$. (B) (i) Representative confocal z-projections depicting autophagosomes using Atg8a-mCherry in the salivary glands from the genotypes a. AB1>Atg8a-mCherry, b. AB1>Atg8a-mCherry; Vps34 RNAi. (ii) Graph representing measurement of Atg8a punctae numbers per unit cell area in the salivary glands from wandering third instar larvae of AB1>Atg8a-mCherry (N=7, n =42), AB1>Atg8a-mCherry ; Vps34 RNAi (N=7, n =42). Student's unpaired t-test with Welch correction showed p value<0.0001. (iii) Representative confocal z-projections depicting autophagosomes using Atg8a-mCherry in the salivary glands from the genotypes a. AB1>Atg8a-mCherry, b. AB1>Atg8a-mCherry; UVRAG RNAi. (iv) Graph representing measurement of Atg8a punctae numbers per unit cell area in the salivary glands from wandering third instar larvae of AB1>Atg8a-mCherry (N=5, n =30), AB1>Atg8a-mCherry; UVRAG RNAi (N=5, n =33). Mann-Whitney test with showed p value = 0.0126. (v) Representative confocal z-projections depicting autophagosomes using Atg8a-mCherry in the salivary glands from the genotypes a. AB1>Atg8a-mCherry, b. AB1>Atg8a-mCherry; Atg1 RNAi. (vi) Graph representing measurement of Atg8a punctae numbers per unit cell area in the salivary glands from wandering third instar larvae of AB1>Atg8a-mCherry (N=6, n =30), AB1>Atg8a-mCherry ; Atg1 RNAi (N =5 , n =20). Student's unpaired t-test with Welch correction showed p value<0.0001. Scale bar indicated at 20 μ m for all the representative images

Minor Comments:

Fig 1A: this is a slightly confusing diagram and could perhaps be made a little clearer. For an example, the arrows are not clearly differentiating phosphorylation from dephosphorylation events. Also, the choice of colour for the phosphatase arrows (brown-red) and kinases (also appearing brown-red) makes it harder to follow this figure.

Similar comment applies to S4B: PI could be depicted as an unphosphorylated version of PI3P/PI5P and drawn in the centre.

Response: Thank you for pointing this out. We have modified the figures according to the suggestions.

Line 301: "lipidated Atg8a fuses with the formed omegasome" Atg8a fusion with omegasome is not an accurate description of the early autophagosome biogenesis events.

Response: Thank you for pointing this out. We have modified this sentence according to the suggestions.

A new image (similar to Fig 1A) depicting how PIP4K affect PI3P levels to summarise the findings of this manuscript would be helpful.

Response: We have constructed a summary diagram in our preliminary revision. This is now new Figure 8.

Figure R7: Schematic showing the regulation of PI3P at a autophagic membranes downstream of Atg1. In this study, dPIP4K which metabolizes PI3P has been shown to genetically interact with Vps34 and Mtm, two established PI3P regulators. Atg8a marked autophagosomes mature to autolysosomes and functions in maintaining normal cell size in wild type cells. In *dPIP4K²⁹* cells, PI3P levels increase, leading to an increase in Atg8a punctae by a combination of increased autophagy initiation and possible defects in maturation, which in turn leads to decreased cell size phenotype.

The material and methods is an important section in this paper: a more thorough description of the methods, especially those referred to previous publications would be very helpful. The authors can at least add a brief outline of the methods they followed and include contents of buffers used.

Response: We have provided an in-depth outline of the methods.

Concern 1 is about the level of PIP₂/PI_{4,5}P₂, the product of PIP4K, in the dPIP4K²⁹ model. This was not measured in the study. The authors claim page 5 that: "This observation suggests that the ability of dPIP4K to regulate cell size does not depend on the pool of PI(4,5)P₂ that it generates... based on the fact that re-expression a mutation that hPIP4Kβ[A381E] in the salivary glands of dPIP4K²⁹ (AB1>hPIP4Kβ[A381E]; dPIP4K²⁹) (Figure S1A) did not rescue the reduced cell size. This mutation hPIP4Kβ[A381E] was generated in a study by Kunz et al. (2002) where they demonstrated by in vitro kinase assay that it cannot utilize PI5P as a substrate but can produce PI(4,5)P₂ using PI4P as a substrate. In the same study, using MG-63 cells, Kunz et al. propose that the A381E mutation did not metabolize PI5P as it lost its plasma membrane localization. In my opinion the author should strength their claim about the role of dPIP4K independently of PI(4,5)P₂ by addressing the level of PI(4,5)P₂ in their model biochemically by mass spectrometry as they have this powerful tool and support this by using PH-PLCd probe to detect PI(4,5)P₂. Also, as they use completely different model as Kunz et al. they should verify, if possible, the localization of hPIP4Kβ [A381E] vs WT PIP4Kβ in salivary glands.

Response - We would like to draw the attention of the reviewer to previous work from our group that PI(4,5)P₂ levels are not reduced in dPIP4K mutants.

- (i) In our first study of dPIP4K we have measured the total level of PI(4,5)P₂ using biochemical techniques and it is not decreased in dPIP4K mutants (Figure 2F in (Gupta et al., 2013)
- (ii) In a subsequent study (Sharma et al., 2019) using a PI(4,5)P₂ probe (PH-PLCδ::GFP), we have shown that there is no reduction in the level of PI(4,5)P₂ at the plasma membrane of salivary glands in dPIP4K mutants.

We have carried out experiments to determine the localization of wild type PIP4K2B and hPIP4K2B [A381E] in the *Drosophila* salivary glands. We confirmed equivalent protein expression of the GFP tagged WT PIP4K2B and hPIP4K2B [A381E] in transiently transfected S2R+ cells (Figure R8A). Figure R8B depicts the similarity in GFP fluorescence (in green) between the two constructs in S2R+ cells. In salivary glands expressing the two constructs, we observed similar expression pattern between the two constructs in the cytosol and a significant proportion of the signal coming from the nucleus (Figure R8C).

Figure R8: (A) Immunoblot from *Drosophila* S2R+ cells showing the expression of hPIP4K2B::eGFP and hPIP4K2B A381E::eGFP. Actin was used as the loading control. (B) Representative confocal z-projections depicting hPIP4K2B::eGFP and hPIP4K2B A381E::eGFP localisation in *Drosophila* S2R+ cells. Scale bar indicated at 5 µm. (C) Representative confocal z-projections depicting hPIP4K2B::eGFP and hPIP4K2B A381E::eGFP localisation in *Drosophila* salivary glands. Scale bar indicated at 20 µm.

Concern

3:

Page 10: "we tagged dPIP4K with the tandem FYVE domain at the C-terminus end of the protein (dPIP4K2XFYVE) to target it to the PI3P enriched endosomal compartment and reconstituted this in the background of dPIP4K29. We did not observe a significant change in the cell size of dPIP4K29" I really don't understand the relevance of this experiment. FYVE tandem will bind to PI3P whenever it was in the cell (Lysosomes, autophagosome). Why the authors claim that the expression of restricted dPIP4K2XFYVE will be restricted to the endosomes. I think that this experiment is confusing and should be removed.

Response: Our approach for this experiment was based on the understanding that 2XFYVE would target dPIP4K exclusively to the early endosomes. We would thank Reviewer #3 for raising this concern. In response to another comment relating to whether 2XFYVE also localizes to autophagosomes, we have provided experimental evidence that this domain does not localize a tagged protein to autophagosomes, rather only to early endosomes (Figure R6).

- Should the authors qualify some of their claims as preliminary or speculative, or remove them altogether?

See concern 1 to 3.

- Would additional experiments be essential to support the claims of the paper? Request additional experiments only where necessary for the paper as it is, and do not ask authors to open new lines of experimentation.

Yes, the proposed experiments in concern 1-3 are not difficult to address as the authors have all the appropriate tools to manage this.

- Are the suggested experiments realistic in terms of time and resources? It would help if you could add an estimated cost and time investment for substantial experiments.

Yes. It is not time consuming and not costly according to their expertise, available tools and materials that they used through the study.

- Are the data and the methods presented in such a way that they can be reproduced?

Yes

- Are the experiments adequately replicated and statistical analysis adequate?

Yes

Minor comments:

- Specific experimental issues that are easily addressable.

References

Goulden BD, Pacheco J, Dull A, Zewe JP, Deiters A, Hammond GRV. 2019. A high-avidity biosensor reveals plasma membrane PI(3,4)P2 is predominantly a class I PI3K signaling product. *J Cell Biol* 218:1066–1079. doi:10.1083/JCB.201809026

Gupta A, Toscano S, Trivedi D, Jones DR, Mathre S, Clarke JH, Divecha N, Raghu P. 2013. Phosphatidylinositol 5-phosphate 4-kinase (PIP4K) regulates TOR signaling and cell growth during Drosophila development. *Proc Natl Acad Sci U S A* 110:5963–5968. doi:10.1073/pnas.1219333110

Kunz J, Fuelling A, Kolbe L, Anderson RA. 2002. Stereo-specific substrate recognition by phosphatidylinositol phosphate kinases is swapped by changing a single amino acid residue. *J Biol Chem* 277:5611–9.

Rusten TE, Rodahl LM, Pattni K, Englund C, Samakovlis C, Dove S, Brech A, Stenmark H. 2006. Fab1 phosphatidylinositol 3-phosphate 5-kinase controls trafficking but not silencing of endocytosed receptors. *Mol Biol Cell* 17:3989–4001. doi:10.1091/mbc.E06-03-0239

Sharma S, Mathre S, Ramya V, Shinde D, Raghu P. 2019. Phosphatidylinositol 5 Phosphate 4-Kinase Regulates Plasma-Membrane PIP3 Turnover and Insulin Signaling. *Cell Rep* 27:1979-1990.e7. doi:10.1016/j.celrep.2019.04.084

2. Description of the revisions that have already been incorporated in the transferred manuscript

Please insert a point-by-point reply describing the revisions that were already carried out and included in the transferred manuscript. If no revisions have been carried out yet, please leave this section empty.

We have presently not made any revisions to the manuscript originally submitted to Review Commons. These will be done as outlined above as we go forward, in consultation with journals who consider the manuscript for publication following revision.

3. Description of analyses that authors prefer not to carry out

Please include a point-by-point response explaining why some of the requested data or additional analyses might not be necessary or cannot be provided within the scope of a revision. This can be due to time or resource limitations or in case of disagreement about the necessity of such additional data given the scope of the study. Please leave empty if not applicable.

Concern

2:

Page 7: The author used Mtm tagged constructs (mCherry and GFP) and measure its phosphatase activity toward PI(3,5)P2 and they did not show any obvious activity. I would like to suggest the use of

untagged (or small tag construct, Flag or HA) for the expression experiment in S2R+ cell as it is known that active myotubularins in other cell model as well as in vitro have a strong 3-phosphatase activity toward PI(3,5)P₂. By looking at the graph FigureS2 Bii, we could clearly see a big disparity within mCherry-Mtm data points. This experiment should be more strengthened by additional experimental points but also by using a positive CTRL where PI(3,5)P₂ level drops (inhibition of PIKfyve by Apilimod).

Response: We thank Reviewer #3 for this comment. We do agree that other cell models have detected strong 3-phosphatase activity on PI(3,5)P₂. Although, in our studies we observe that C-terminal tagged version of Mtm is active on PI3P, thus ruling out a possibility of the enzyme being inactive. We will definitely try to construct a minimally tagged (HA or FLAG-tagged) version of Mtm and perform the 3-phosphatase activity assay. However, we would like to state that for the purposes of this study, Mtm-GFP which is active on PI3P only suits as a better molecular tool that specifically affected PI3P levels *in vivo*. Therefore, we believe that our conclusions regarding the relationship between PI3P and cell size would not be affected by the above discrepancy.

May 26, 2023

RE: Life Science Alliance Manuscript #LSA-2023-01920R

Prof. Padinjat Raghu
National Centre for Biological Sciences
Cellular Organization and Signalling
TIFR GKVK Campus
Bangalore, Karnataka 560065
India

Dear Dr. Raghu,

Thank you for submitting your revised manuscript entitled "PI3P dependent regulation of cell size and autophagy by phosphatidylinositol 5-phosphate 4-kinase". We would be happy to publish your paper in Life Science Alliance pending final revisions necessary to meet our formatting guidelines.

- please address Reviewer 2's remaining comments
- please upload your manuscript as a doc file
- please upload both your main and supplementary figures as separate single files
- please add the author contributions and a conflict of interest statement to the main manuscript text
- please use the [10 author names, et al.] format in your references (i.e. limit the author names to the first 10)
- please add a scale bar to Figure 5G
- you may consider uploading Figure 8 as a Graphical Abstract instead, but this is up to you

A. FINAL FILES:

B. MANUSCRIPT ORGANIZATION AND FORMATTING:

Thank you for your attention to these final processing requirements. Please revise and format the manuscript and upload materials within 5 days.

Sincerely,

Reviewer #2 (Comments to the Authors (Required)):

The authors utilise a drosophila model to investigate the molecular mechanisms underlying the role of dPIP4K in regulating cell size. They suggest that dPIP4K can regulate PI3P levels, which through enhancing autophagy, results in a reduction in cell size. Overall, this is an interesting study. The authors have attempted to address most of my initial comments and I only have the following minor comments:

- The authors have adjusted their discussions to include various mechanisms through which PIP4K regulates PI3P levels, which is sufficient at this stage of the manuscript.

- Figure 4E: I still think it is inaccurate to say that none of the phosphatases transcripts are changed in the text when the statistical analyses shows a significant difference. The values in the graph are low so it is difficult to appreciate the fold change. The text related to this figure should still be adjusted to more accurately describe the data.

- It is unfortunate that the authors could not get the ATG14L knockdown to work. The findings using UVRAG knockdown are interesting even though the degree of knockdown seems to be comparative to ATG14L knockdown. For the authors' reference and as stated in my initial comments, expressing FYVE domains in live cells can be inhibitory and change vesicle dynamics. These probes have been frequently used to stain fixed cells and can be used to look at both autophagosome and endosome PI3P (PMID:28813193).

Reviewer #3 (Comments to the Authors (Required)):

This study by Ghosh et al. proposes a role for phosphatidylinositol 5-phosphate 4-kinase (PIP4K) in regulating PI3P levels in vivo. They use loss-of-function Drosophila model of the only PIP4K gene (dPIP4K29) to investigate the PI3P and PI(3,5)P2 metabolizing enzymes. First, they showed that loss of function of PIP4K leads to reduced cell size in larval salivary glands and this was attributed to the elevated level of PI3P. Then, they modulated enzymes involved in PI3P metabolism (kinases and phosphatases) and propose the implication of the PI3P phosphatase myotubularin (Mtm) and the Pi3k Class III (PI3K59F) in PIP4K-dependent cell size control. Finally, as PI3P has an established role in autophagy, they modulate the autophagy related gene (atg1) and connect the observed increase of PI3P level to the upregulation of autophagy in dPIP4K29 model. The authors used genetic manipulations of dPIP4K29 models as well as specialized lipidomic expertise (phosphoinositide measurement using mass spectrometry and PI-kinase/phosphatase assays) to address their conclusions. The experimental strategies were well designed and major conclusions were in line with the obtained results. The work provides a significant advance in understanding the PIPs conversion mechanisms within specific organelles such as autophagosomes and how this impact cell shape.

The authors revision addressed the majority of reviewer's comments.
No additional issues

Response to Editor's comments

Along with points mentioned below, please tend to the following:
please address Reviewer 2's remaining comments

-please upload your manuscript as a doc file

This has been done

-please upload both your main and supplementary figures as separate single files

This has been done

-please add the author contributions and a conflict of interest statement to the main manuscript text

This has been done

-please use the [10 author names, et al.] format in your references (i.e. limit the author names to the first 10)

This has been done and corrected where appropriate.

-please add a scale bar to Figure 5G

This has been done

-you may consider uploading Figure 8 as a Graphical Abstract instead, but this is up to you

This has been done.

Reviewer #2 (Comments to the Authors)

The authors utilise a drosophila model to investigate the molecular mechanisms underlying the role of dPIP4K in regulating cell size. They suggest that dPIP4K can regulate PI3P levels, which through enhancing autophagy, results in a reduction in cell size. Overall, this is an interesting study. The authors have attempted to address most of my initial comments and I only have the following minor comments:

- The authors have adjusted their discussions to include various mechanisms through which PIP4K regulates PI3P levels, which is sufficient at this stage of the manuscript.

- Figure 4E: I still think it is inaccurate to say that none of the phosphatases transcripts are changed in the text when the statistical analyses shows a significant difference. The values in the graph are low so it is difficult to appreciate the fold change. The text related to this figure should still be adjusted to more accurately describe the data.

We have adjusted the text for this comment in the main manuscript as follows:

However we observed that the transcript levels of *mtm* and *CG3530* – a putative 3-phosphatase were decreased in *dPIP4K*²⁹ as compared to controls (Figure 4E), which does not directly correlate with the lack of decrease in total 3-phosphatase activity observed in Figure 4D.

- It is unfortunate that the authors could not get the ATG14L knockdown to work. The findings

using UVRAG knockdown are interesting even though the degree of knockdown seems to be comparative to ATG14L knockdown. For the authors' reference and as stated in my initial comments, expressing FYVE domains in live cells can be inhibitory and change vesicle dynamics. These probes have been frequently used to stain fixed cells and can be used to look at both autophagosome and endosome PI3P (PMID:28813193).

We have noted the reviewer comments and also the supplied reference PMID:28813193 and thank them for the same. However we are also aware, through discussions with members of the autophagy community that despite such published examples of the use of the FYVE domain to monitor both the endosomal and autophagic pools of PI3P, there is not broad agreement in the autophagy community that the FYVE domain can be used to monitor the autophagic pool of PI3P. In the absence of such broad agreement we have refrained from making a conclusion on the autophagic pool of PI3P using the FYVE domain.

Although the knockdown as measured by transcript levels of both UVRAG and ATG14L may have been equivalent, we are confident that in the case of UVRAG knockdown the ca. 50% transcript level has functional consequences on PI3P levels as the number of FYVE punctae was reduced. However, in the absence of a unequivocal method to monitor autophagosomal PI3P levels we cannot be sure if there is sufficient depletion of PI3P levels on ATG14L depletion in our experiment.

June 2, 2023

RE: Life Science Alliance Manuscript #LSA-2023-01920RR

Prof. Padinjat Raghu
National Centre for Biological Sciences
Cellular Organization and Signalling
TIFR GKVK Campus
Bangalore, Karnataka 560065
India

Dear Dr. Raghu,

Thank you for submitting your Research Article entitled "PI3P dependent regulation of cell size and autophagy by phosphatidylinositol 5-phosphate 4-kinase". It is a pleasure to let you know that your manuscript is now accepted for publication in Life Science Alliance. Congratulations on this interesting work.

DISTRIBUTION OF MATERIALS:

Again, congratulations on a very nice paper. I hope you found the review process to be constructive and are pleased with how the manuscript was handled editorially. We look forward to future exciting submissions from your lab.

Sincerely,
